# Rectified Robust Policy Optimization for Model-Uncertain Constrained Reinforcement Learning without Strong Duality

**Shaocong Ma**                                                   *scma0908@umd.edu*
*Department of Computer Science*
*University of Maryland, College Park*

**Ziyi Chen**                                                         *zc286@umd.edu*
*Department of Computer Science*
*University of Maryland, College Park*

**Yi Zhou**                                                         *yi.zhou@tamu.edu*
*Department of Computer Science and Engineering*
*Texas A&M University*

**Heng Huang**                                                       *heng@umd.edu*
*Department of Computer Science*
*University of Maryland, College Park*

**Reviewed on OpenReview:** *https://openreview.net/forum?id=7l63xwAgAW*

## Abstract

The goal of robust constrained reinforcement learning (RL) is to optimize an agent's performance under the worst-case model uncertainty while satisfying safety or resource constraints. In this paper, we demonstrate that strong duality does not generally hold in robust constrained RL, indicating that traditional primal-dual methods may fail to find optimal feasible policies. To overcome this limitation, we propose a novel primal-only algorithm called Rectified Robust Policy Optimization (RRPO), which operates directly on the primal problem without relying on dual formulations. We provide theoretical convergence guarantees under mild regularity assumptions, showing convergence to an approximately optimal feasible policy with iteration complexity matching the best-known lower bound when the uncertainty set diameter is controlled in a specific level. Empirical results in a grid-world environment validate the effectiveness of our approach, demonstrating that RRPO achieves robust and safe performance under model uncertainties while the non-robust method can violate the worst-case safety constraints.

## 1 Introduction

In many practical reinforcement learning (RL) applications, it is critical for an agent to not only maximize expected cumulative rewards but also satisfy certain constraints, such as safety requirements (Yao et al., 2024; Gu et al., 2024b) or resource limitations (Wang et al., 2023c). However, real-world environments often diverge from the training environment due to model mismatch (Roy et al., 2017; Viano et al., 2021; Zhai et al., 2024; Wang et al., 2024) and environment uncertainty (Lütjens et al., 2019; Wang & Zou, 2021; Ma et al., 2023). Such discrepancies can lead to significant performance degradation and, more severely, violations of constraints, which is unacceptable in safety-critical applications. For instance, an autonomous robot may encounter unforeseen transitions due to equipment aging or mechanical failures: In the environment navigation and the robust control tasks, the robot make take the action that perform better in the ideal simulated environment due to the ideal equipment condition but suffer high penalty from higher energy cost or unexpected operations in the worst-case scenario.

Despite its practical importance, robust constrained RL has been relatively underexplored in the literature. Two closely related areas are robust RL (Bagnell et al., 2001; Nilim & Ghaoui, 2005; Iyengar, 2005) and constrained RL (Altman, 1999; Wachi & Sui, 2020). Robust RL focuses on optimizing performance under model uncertainties but typically does not consider constraints. Constrained RL aims to optimize performance while satisfying certain constraints but often assumes a fixed environment without uncertainties. Seamlessly combining two fields presents inherent challenges.

To address these challenges, we propose a framework for robust constrained RL under model uncertainty. Specifically, we consider Markov Decision Processes (MDPs) where the transition dynamics are not fixed but lie within an uncertainty set, which is commonly known as the robust MDPs (Mannor et al., 2016; Ho et al., 2018; Tamar et al., 2013; Grand-Clément & Kroer, 2021). Our objective is to optimize the worst-case cumulative reward over this uncertainty set while ensuring that all constraints are also simultaneously satisfied in the worst-case scenario. This robust approach ensures that the agent's policy remains effective and safe even when the environment deviates from the nominal model.

A common approach to solving such constrained problems is the primal-dual method (Altman, 1999; Paternain et al., 2019; Bai et al., 2022; Liang et al., 2018; Chen & Wang, 2016; Mahadevan et al., 2014; Chen et al., 2022), which leverages the strong duality property to efficiently find optimal policies. Strong duality allows the original constrained problem to be solved by considering its dual problem, simplifying computations and enabling convergence guarantees. However, a crucial question arises:

> ***Q1:*** *Does strong duality hold in robust constrained RL?*

In this paper, we address this question head-on. We first demonstrate that, unfortunately, **strong duality does not generally hold in robust constrained RL**. The presence of model uncertainties breaks the Fenchel-Moreau condition, the common routine of showing the strong duality in the non-robust constrained RL (Altman, 1999). We construct a specific counterexample where the duality gap, the difference between the optimal values of the primal and dual problems, is strictly positive. This finding indicates that traditional constrained RL algorithm may fail to be directly generalized to the robust constrained setting; we also note that the non-zero duality gap does not exclude the potential global convergence of primal-dual algorithm. Recognizing this fundamental issue, we are motivated to ask the following question:

> ***Q2:*** *Can we develop a non-primal-dual algorithm for solving robust constrained RL problems with provable convergence guarantees?*

To address this question, we introduce the Rectified Robust Policy Optimization (RRPO), a primal-only algorithm adapted from the CRPO (Xu et al., 2021). RRPO is specifically designed for robust constrained RL, which bypasses the issues associated with the duality gap. Our algorithm operates directly on the primal problem, ensuring constraint satisfaction and robustness without relying on dual formulations or strong duality assumptions. We summarize our key contributions as follows:

1. **Counterexample—Non-Zero Duality Gap:** In Section 3, we provide a concrete example showing that strong duality does not hold in robust constrained RL. To the best of our knowledge, this negative result is the **first** theoretical example showing the non-zero duality gap in the existing literature, which resolves an open problem in constrained robust RL. We emphasize that establishing such a counterexample is nontrivial and crucial for understanding the theoretical boundaries of constrained robust RL.

2. **Proposed Primal-Only Algorithm—RRPO:** Motivated by the lack of strong duality of constrained robust RL problems, we introduce RRPO, a primal-only algorithm designed to solve robust constrained RL problems without relying on strong duality. Moreover, in Section 4, we rigorously analyze the convergence properties of RRPO. Specifically, we prove that under appropriate conditions, RRPO converges to an approximately optimal feasible policy $\pi^*$ within a specified tolerance $\delta$ in the worst-case scenario. When the uncertainty set is sufficiently small, our derived convergence rate and iteration complexity also **recover the best-possible lower bound** for non-robust constrained RL problems (Vaswani et al., 2022).

3. **Empirical Validation:** We validate the effectiveness of RRPO through experiments in a grid-world environment and the classical mountain car environment. Our results show that RRPO achieves robust

and safe performance under model uncertainties, outperforming the original CRPO method that may fail to maintain constraint satisfaction in the worst-case scenario.

## 1.1 Related Work

**Robust Constrained RL**  Here, we mainly explore the existing literature regarding robust constrained RL. In Appendix A.1, we provide other related work. Robust constrained RL considers the problem of optimizing performance while satisfying constraints in the worst-case scenario. Although robust RL and constrained RL have each been extensively studied, fewer works address their intersection. In Russel & Petrik (2020), the authors investigate robust constrained RL and propose a heuristic approach that estimates robust value functions and employs a standard policy gradient method (Sutton et al., 1999), substituting the nominal value function with the robust one. However, as Wang et al. (2022) points out, this approach overlooks how the worst-case transition kernel depends on the policy, resulting in updates that do not correspond to actual gradients of the robust value function and thus lack theoretical convergence guarantees. To remedy this, Wang et al. (2022) introduces a robust primal-dual algorithm for solving robust constrained RL problems. However, this method assumes the strong duality, which we will show later, generally does not hold in robust constrained RL. Several other studies also examine the strong duality of robust constrained RL problems: Ghosh (2024) points out that the standard routine in proving the strong duality of constrained RL problems (Panaganti & Kalathil, 2022) cannot hold in the robust case. Zhang et al. (2024) proves the strong duality by considering a different policy space, which is different from the space considered in Paternain et al. (2019). We include further discussion on it in Appendix A.2. Complementing these works, He et al. (2025) tackle the online distributionally robust off-dynamics setting, and present the ORBIT algorithm that attains sublinear regret with matching upper and lower sample-complexity bounds under general $f$-divergence uncertainty sets.

The uncertainty set captures the discrepancy between training and deployment environments. In our work, we mainly focus on the $(s, a)$-rectangular uncertainty set (Iyengar, 2005; Nilim & Ghaoui, 2005), where the transition uncertainty is modeled independently for each state-action pair. As shown by Lu et al. (2024); Kumar et al. (2023); Wang et al. (2023a); Wang & Zou (2021), this structure induced by a specific norm or divergence usually enables efficient solution methods through robust value iteration and linear programming, preserving the tractability of dynamic programming. While our work is also potential to be extended to other settings when a valid robust policy evaluation algorithm presents, including:

**$s$-Uncertainty Sets**  The $s$-rectangular uncertainty set (Li et al., 2022; Li & Shapiro, 2023; Kumar et al., 2023) relaxes independence by allowing nature to choose transitions jointly across all actions at a given state. The widely known results include the robust policy gradient formula (Li et al., 2022) and the closed-form solution for $p$-norm uncertainty sets (Kumar et al., 2023). These results make it possible to solve the robust value function more efficiently.

**$d$-Uncertainty Sets**  More recently, the $d$-rectangular uncertainty set (Ma et al., 2022) has been proposed for linear MDPs to further generalize the existing concepts of $(s, a)$-rectangular uncertainty sets, where the ambiguity is structured in a low-dimensional decision space, enabling scalable robust value iteration in high-dimensional settings. Representative works include Blanchet et al. (2023); Liu & Xu (2024a); Liu et al. (2024); Liu & Xu (2025). Remarkably, this setting has also been widely explored in the offline RL setting (Tang et al., 2024; Liu & Xu, 2024b). These results make it possible to extend our results potentially to be able to handle more complicated robust MDPs with the function approximation.

Additionally, we discuss our connection to other setting in the robust reinforcement learning and constrained reinforcement learning.

**Robust RL with Reward and State Uncertainty**  Modern robust RL tackles misspecified or learned rewards by optimizing worst-case or risk-aware returns. Early formulations blending nominal and worst-case objectives (Xu & Mannor, 2006; Delage & Mannor, 2010) have been extended by coupling-aware uncertainty sets that avoid excessive conservatism (Mannor et al., 2012). Recent work uses Bayesian reward ensembles and CVaR objectives for preference-based or RLHF settings, giving reliability under noisy learned rewards

(Brown et al., 2020). Newest advances address uncertainty *inside* the reward or state signal itself. Seeing-is-Not-Believing RSC-MDPs explicitly model spurious correlations in the *state* and learn policies robust to such confounding (Ding et al., 2023). Mirror-Descent inverse RL adds robustness directly to the *learned reward* via adversarial mirror steps with $O(1/T)$ regret (Han et al., 2022). Distributional Reward Estimation builds a reward-uncertainty distribution whose risk-sensitive aggregation stabilises multi-agent training (Hu et al., 2022). Moreover, robust MDPs cast dynamics error as a zero-sum game; rectangular ambiguity sets admit efficient robust DP with deterministic optimal policies (Iyengar, 2005; Nilim & Ghaoui, 2005). Follow-ups loosen rectangularity to retain tractability while reducing pessimism (Wiesemann et al., 2013; Goyal & Grand-Clement, 2023). Practical algorithms combine adversarial disturbances or domain randomization with policy learning to harden policies against dynamics shifts (Pinto et al., 2017).

**RL with Hard Constraints**   CMDP theory guarantees optimal stationary policies and LP or primal–dual solutions (Altman, 1999; Borkar, 2005). Deep RL implementations such as CPO (Achiam et al., 2017) and RCPO (Tessler et al., 2019) enforce budget or safety limits during learning. Runtime *shielding* synthesised from temporal-logic specifications now guarantees safety even under partial observability (Carr et al., 2023). Quantile-Constrained RL enforces outage-probability bounds by constraining the cost distribution's tail rather than its expectation (Jung et al., 2022). Trust-Region Safe Distributional Actor–Critic simultaneously handles multiple constraints with distributional critics and provable convergence (Kim et al., 2023). Model-predictive safety filters based on learned control-barrier functions enforce hard state constraints in unknown stochastic systems while optimizing performance (Wang et al., 2023b). Together these methods wrap (or jointly train) base policies to *guarantee* constraint satisfaction without relying on soft penalties.

**Safe RL**   Safe exploration confines learning to recoverable states (Moldovan & Abbeel, 2012) or certified Lyapunov safe sets (Berkenkamp et al., 2017). Action-level intervention layers correct unsafe commands on the fly (Dalal et al., 2018a), while risk-sensitive criteria such as CVaR directly penalize catastrophic tails (Chow et al., 2018a). These threads, alongside recent hard-constraint and shielding techniques above, provide complementary, constraint-respecting alternatives that our revised Related-Work section now covers.

**Robust Constrained Multi-Agent Systems**   An important extension of robust constrained RL is its application to multi-agent settings. The concept of robustness has been generalized to multi-agent reinforcement learning (MARL) (Zhang et al., 2020; Ma et al., 2023; Jiao & Li, 2024), particularly in addressing state uncertainty (He et al., 2023a; Han et al., 2024). Notably, He et al. (2023b) were the first to jointly consider robustness and constraints in a real-world scenario, proposing the ROCOMA algorithm, which achieves efficient EV rebalancing under transition kernel uncertainty. Many topics in constrained robust MARL remains unexplored, making it a promising direction with significant potential for further research.

## 2   Preliminaries and Problem Formulation

In this section, we formalize our problem setting in the context of online robust constrained RL, where the transition probabilities are unknown. We re-use an existing robust policy evaluation algorithm to estimate the approximate value function.

### 2.1   Robust MDPs

A *Robust Markov Decision Process* (Robust MDP) is defined by the tuple $(\mathcal{S}, \mathcal{A}, \mathcal{P}, r, \gamma)$, where $\mathcal{S}$ is a finite state space, $\mathcal{A}$ is a finite action space, $\mathcal{P}$ represents the uncertainty set of transition probabilities with $\Delta(\mathcal{S})$ denoting the probability simplex over $\mathcal{S}$, $r : \mathcal{S} \times \mathcal{A} \to \mathbb{R}$ is the reward function, $\gamma \in [0, 1)$ is the discount factor. We denote $\mu \in \Delta(\mathcal{S})$ as the initial state distribution.

In an robust MDP, the transition probabilities are not fixed but belong to an uncertainty set; usually, the uncertainty set $\mathcal{P}$ is defined as the *s*-rectangular set (Derman et al., 2021; Wang et al., 2023a; Wiesemann et al., 2013; Kumar et al., 2023)

$$\mathcal{P} := \times_{s \in \mathcal{S}} \mathcal{P}_s,$$

or $(s, a)$-rectangular set (Wiesemann et al., 2013; Kumar et al., 2023)

$$\mathcal{P} := \times_{(s,a) \in \mathcal{S} \times \mathcal{A}} \mathcal{P}_{(s,a)}.$$

Here, instead of assuming a specific type of uncertainty set as in many existing literature (Wang & Zou, 2021; Wang et al., 2022), we work on general uncertainty sets but simply assume that the robust value function over these uncertainty set is computationally available. Notably, for many well-known uncertainty sets, such as the *p*-norm (Kumar et al., 2023), IPM (Zhou et al., 2024), and R-contamination (Wang & Zou, 2021) uncertainty set, the robust value function can be efficiently calculated without hurting the sample complexity.

Let the policy $\pi : \mathcal{S} \to \Delta(\mathcal{A})$ map each state to a probability distribution over actions. In robust RL, the robust value function $V^{\pi}(s)$ under policy $\pi$ starting from state $s$ is defined as the worst-case expected discounted cumulative reward:

$$V^{\pi}(s) = \inf_{P \in \mathcal{P}} \mathbb{E}_{\pi, P} \left[ \sum_{t=0}^{\infty} \gamma^t r(s_t, a_t) \, \middle| \, s_0 = s \right],$$

where the expectation is taken over the trajectories generated by following policy $\pi$, with $a_t \sim \pi(\cdot \mid s_t)$ and $s_{t+1} \sim P(\cdot \mid s_t, a_t)$ for $P \in \mathcal{P}$. The objective is to find an optimal policy $\pi^*$ that maximizes the worst-case expected cumulative reward from the initial state distribution $\mu$:

$$V^{\pi^*}(\mu) = \max_{\pi} V^{\pi}(\mu),$$

where $V^{\pi}(\mu) := \mathbb{E}_{s \sim \mu}[V^{\pi}(s)]$.

## 2.2 Robust Constrained MDPs

In many applications, it is essential to optimize the reward while satisfying certain constraints, even under model uncertainty. *Constrained robust MDPs* (Wang et al., 2022; Zhang et al., 2024; Sun et al., 2024; Ghosh, 2024) extend the robust MDP framework by incorporating multiple constraints.

Let there be $I$ constraint reward functions $r_i : \mathcal{S} \times \mathcal{A} \to \mathbb{R}$ for $i = 1, 2, \ldots, I$. The robust expected cumulative reward under policy $\pi$ for constraint $i$ is given by:

$$V_i^{\pi}(s) = \inf_{P \in \mathcal{P}} \mathbb{E}_{\pi, P} \left[ \sum_{t=0}^{\infty} \gamma^t r_i(s_t, a_t) \, \middle| \, s_0 = s \right],$$

and $V_i^{\pi}(\mu) = \mathbb{E}_{s \sim \mu}[V_i^{\pi}(s)]$ is the robust expected cumulative cost from the initial distribution $\mu$.

The constrained robust MDP aims to find a policy that maximizes the worst-case reward while ensuring that each constraint is satisfied under the worst-case transition dynamics:

$$\begin{aligned} \max_{\pi} \quad & V_0^{\pi}(\mu) \\ \text{s.t.} \quad & V_i^{\pi}(\mu) \geq d_i, \quad \text{for } i = 1, 2, \ldots, I, \end{aligned} \tag{1}$$

where $V_0^{\pi}(\mu)$ denotes the robust expected cumulative reward, and $d_i$ are the specified thresholds for the constraints. That is, a constrained robust MDP is defined by the tuple $(\mathcal{S}, \mathcal{A}, \mathcal{P}, \{r_i\}_{i=0}^{I}, \{d_i\}_{i=1}^{I}, \gamma)$, where $\{r_i\}_{i=0}^{I}$ and $\{d_i\}_{i=1}^{I}$ extend the original robust MDP to include these constraint reward function $r_i$ and the threshold $d_i$.

## 2.3 Duality Gap of Robust Constrained MDPs

In constrained optimization, the concept of duality plays a pivotal role in formulating and solving problems (Boyd & Vandenberghe, 2004; Bertsekas et al., 2003). The *duality gap* is the difference between the optimal values of the primal problem and its dual. When this gap is zero, we say that *strong duality* holds, allowing the

primal and dual problems to have the same optimal value. This property is instrumental in many optimization algorithms, particularly in convex optimization, where it enables efficient computation of optimal solutions via dual methods. For the constrained robust MDP defined earlier, we incorporate the constraints into the optimization objective, formulating the *Lagrangian* of the constrained robust RMDP. The Lagrangian combines the objective function and the constraints using Lagrange multipliers $\lambda = (\lambda_1, \lambda_2, \ldots, \lambda_I) \geq 0$:

$$\mathcal{L}(\pi, \lambda) = V_0^\pi(\mu) - \sum_{i=1}^I \lambda_i \left(d_i - V_i^\pi(\mu)\right). \tag{2}$$

In this formulation, $\mathcal{L}(\pi, \lambda)$ is the Lagrangian function, and $\lambda_i \geq 0$ are the Lagrange multipliers associated with the constraints. The *primal problem* is defined by maximizing over $\pi$, after minimizing the Lagrangian over $\lambda \geq 0$. That is,

$$\max_\pi \min_{\lambda \geq 0} \mathcal{L}(\pi, \lambda). \tag{3}$$

The *dual problem* is then obtained by minimizing the Lagrangian over $\lambda \geq 0$, after maximizing over $\pi$. Specifically, the dual problem is:

$$\min_{\lambda \geq 0} \max_\pi \mathcal{L}(\pi, \lambda). \tag{4}$$

The *duality gap* $\mathscr{D}$ is defined as the difference between the optimal value of the primal problem and the optimal value of the dual problem.

**Definition 2.1** (Duality gap of robust constrained MDPs). Let $\mathcal{M} := (\mathcal{S}, \mathcal{A}, \mathcal{P}, \{r_i\}_{i=0}^I, \{d_i\}_{i=1}^I, \gamma)$ be a robust constrained MDP. The *duality gap* $\mathscr{D}$ of $\mathcal{M}$ is defined as

$$\mathscr{D} := \left[\max_\pi \min_{\lambda \geq 0} \mathcal{L}(\pi, \lambda)\right] - \left[\min_{\lambda \geq 0} \max_\pi \mathcal{L}(\pi, \lambda)\right], \tag{5}$$

where $\mathcal{L}$ is the Lagrangian function of $\mathcal{M}$ defined by Equation (2).

It has been widely known that, in standard constrained MDPs without robustness considerations, under certain regularity conditions, strong duality holds (Altman, 1999). This means that the duality gap $\mathscr{D}$ is zero, and the optimal value of the primal problem equals that of the dual problem. This property allows us to use primal-dual algorithms effectively to find optimal policies that satisfy the constraints. However, in the next section, we will show that the constrained robust MDPs may not have such nice property, which presents a significant challenge for solving constrained robust RL problems.

## 3 Constrained Robust RL Has Non-Zero Duality Gap

In this section, we present a counterexample demonstrating that the duality gap in robust constrained MDPs (Definition 2.1) can be strictly positive.

**Theorem 3.1.** *There exists a constrained robust MDP such that its duality gap is strictly positive.*

Here, we describe the construction of this counterexample. Then we will briefly describe the analysis of the duality gap. The full proof can be found in Appendix B.

### 3.1 Construction of the Counterexample

Consider a simple MDP with two states, $s_0$ and $s_1$, and two actions, $a_0$ and $a_1$, as depicted in Figure 1. The MDP is defined as follows: *(i)* ***Transitions:*** The initial state is $s_0$. From state $s_1$, any action deterministically transitions back to state $s_0$. From state $s_0$, action $a_0$ deterministically remains in $s_0$. From state $s_0$, action $a_1$ transitions to $s_0$ with probability $p$ and to $s_1$ with probability $1 - p$. *(ii)* ***Robustness:*** There is model uncertainty in the transition probability $p$, such that $p \in [\underline{p}, \overline{p}]$, representing the uncertainty set. *(iii)* ***Reward:*** The reward function for the objective is $r_0(s_0) = 1$ and $r_0(s_1) = 0$. The reward function for the constraint is $r_1(s_0) = 0$ and $r_1(s_1) = 1$. *(iv)* ***Constraints:*** The goal is to maximize the expected cumulative reward of $r_0$ while ensuring that the expected cumulative reward of $r_1$ meets a specified threshold $\rho$ under the worst-case transition probabilities.

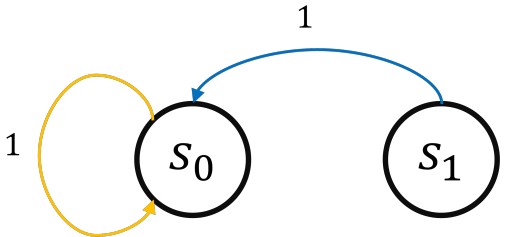
(a) Transitions under action $a_0$

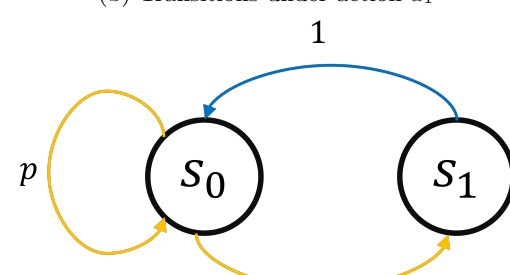
(b) Transitions under action $a_1$

Figure 1: The transition diagram of the MDP considered in Theorem 3.1. At state $s_1$, the agent always moves to state $s_0$ with probability 1, regardless of the action taken. At state $s_0$, the agent has a probability $p$ of staying in the current state when taking action $a_1$, and a probability of 1 of staying in state $s_0$ when taking action $a_0$. The uncertainty only occurs in the transition probability $p$; we let it vary from $[\underline{p}, \overline{p}]$.

### 3.2 Analysis of the Duality Gap

The robust control problem can be formulated as:

$$\max_{\pi} \quad \widetilde{V}_0^{\pi}(s_0) \tag{6}$$

$$\text{s.t.} \quad \widetilde{V}_1^{\pi}(s_0) \geq \rho, \tag{7}$$

where $\widetilde{V}_i^{\pi}(s_0)$ denotes the worst-case value function for reward $r_i$ starting from state $s_0$.

The associated Lagrangian is:

$$\mathcal{L}(\pi, \lambda) = \widetilde{V}_0^{\pi} - \lambda(\rho - \widetilde{V}_1^{\pi}) = \frac{1}{1 - \gamma + \pi_1(1 - \underline{p})(\gamma - \gamma^2)} - \lambda\left(\rho - \frac{\gamma\pi_1(1 - \overline{p})}{1 - \gamma + \pi_1(1 - \overline{p})(\gamma - \gamma^2)}\right).$$

with $\lambda \geq 0$, where $\pi_1 := \pi(a_1|s_0)$.

We proceed to analyze the Lagrangian function and compute the duality gap by evaluating both the primal and dual formulations:

**Primal Problem**  The primal optimization problem given by Equation (3) aims to find the policy $\pi$ that maximizes $\widetilde{V}_0^{\pi}(s_0)$ while satisfying the constraint (7). Here, we directly solve it and obtain

$$\max_{\pi} \min_{\lambda} \mathcal{L}(\pi, \lambda) = \frac{1}{1 - \gamma} - \frac{\rho\dfrac{1 - \underline{p}}{1 - \overline{p}}}{1 - \rho(1 - \gamma) + \rho(1 - \gamma)\dfrac{1 - \underline{p}}{1 - \overline{p}}}.$$

**Dual Problem**  The dual problem given by Equation (4) involves minimizing the Lagrangian over $\lambda \geq 0$ for a fixed policy $\pi$, and then maximizing over $\pi$. The lack of convexity in the robust setting leads to a discrepancy between the solutions obtained from the primal and dual problems.

$$\min_{\lambda} \max_{\pi} \mathcal{L}(\pi, \lambda) = \frac{1}{1 - \gamma} - \frac{1 - \underline{p}}{1 - \overline{p}}\frac{1 + (1 - \overline{p})\gamma}{1 + (1 - \underline{p})\gamma}\rho.$$

It can be obviously observed that when the robustness is absent (i.e. $\underline{p} = \overline{p}$), the primal problem presents the same value as the dual problem.

---

**Algorithm 1:** Rectified Robust Policy Optimization

---

**Input :** initial policy parameters $\theta_0$, empty set $\mathcal{N}_0$

**for** $t = 0, \cdots, T - 1$ **do**
    // Robust Policy Evaluation (e.g. Algorithm 2)
    Evaluate value functions under $\pi_t := \pi_{\theta_t}$: $\hat{Q}_i^{\pi_t}(s, a) \approx Q_i^{\pi_t}(s, a)$ for $i = 0, 1, \ldots, I$ ;
    Sample state-action pairs $(s_j, a_j)$ from the nominal distribution ;
    Compute value estimates $V_i^{\pi_t}$ for $i = 0, \ldots, I$ ;
    **if** $V_i^{\pi_t} \geq d_i - \delta$ *for all* $i = 0, 1, \ldots, I$ **then**
        // Threshold Updates
        Add $\theta_t$ to set $\mathcal{N}_0$ and track the feasible policy achieving the largest value $\pi_{\text{out}} = \pi_t$;
        Update $d_0$: $d_0^{t+1} \leftarrow V_0^{\pi_t}$;
    **else if** $V_i^{\pi_t} < d_i - \delta$ *for some* $i = 1, \ldots, I$ **then**
        // Constraint Rectification
        Maximize $V_i^{\pi_t}$ using Equation (8);
    **else if** $V_0^{\pi_t} < d_0 - \delta$ **then**
        // Objective Rectification
        Maximize $V_0^{\pi_t}$ using Equation (8);

**return:** $\pi_{\text{out}}$

---

**Demonstration of the Duality Gap** By selecting values for the parameters (e.g., $p = 0.5$, $\underline{p} = 0.25$, $\overline{p} = 0.75$, $\gamma = 0.5$, and $\rho = 1$), we can compute the exact values of the duality gap:

$$\mathcal{D} = \max_{\pi} \min_{\lambda \geq 0} \mathcal{L}(\pi, \lambda) - \min_{\lambda \geq 0} \max_{\pi} \mathcal{L}(\pi, \lambda) = \frac{21}{22}.$$

As the result, the strong duality does not generally hold for robust constrained MDPs.

**Implications of a Non-Zero Duality Gap** We have just presented a counterexample showing that strong duality does not generally hold in robust constrained MDPs, which resolves an open problem regarding the strong duality of robust constrained RL problems, highlighting the importance of designing solution methods that do not rely solely on duality. In the subsequent sections, we address these challenges by proposing the primal-only approach, RRPO.

## 4 Solving Robust Constrained RL with Unknown Transition Kernel

The lack of strong duality in robust constrained RL presents significant challenges for traditional primal-dual optimization methods. The presence of a non-zero duality gap means that these methods may fail to find feasible and optimal policies in robust constrained settings. To overcome this obstacle, we develop a primal-only algorithm specifically designed for solving robust constrained RL with unknown transition kernel, which we call RRPO.

### 4.1 Algorithm Design

Given the primal optimization problem:

$$\max_{\pi} \quad V_0^{\pi}(\mu)$$
$$\text{s.t.} \quad V_i^{\pi}(\mu) \geq d_i, \text{ for } i = 1, 2, \ldots I.$$

Here, we note that all $V_i^{\pi}$ $(i = 0, 1, \ldots, I)$ represent the robust value functions; when $i = 0$, we call $V_0^{\pi}$ the objective value function, while when $i \neq 0$, we call $V_i^{\pi}$ the constraint value function. The core concept of the CRPO algorithm (Xu et al., 2021) is to iteratively update the policy by taking gradients with respect

---

**Algorithm 2:** Robust Linear Temporal Difference (Zhou et al., 2024)

---

**Input** : policy $\pi$, number of steps $K$, value function approximation $V_w = \psi^\top w$

**Initialize:** $w_0$, $s_0$;

**for** $k = 0, \ldots, K-1$ **do**

  Sample action $a_k \sim \pi(\cdot \mid s_k)$;

  Sample transition batch $y_{k+1}$ according to the nominal kernel $P_0(\cdot|s_k, a_k)$;

  `// For IPM estimator: (Zhou et al., 2024)`

  Update $w_{k+1} \leftarrow w_k + \alpha_k \psi(s_k)\Big[\big(r(s_k, a_k) + \gamma V_{w_k}(y_{k+1}) - \gamma \delta \|w_{k,2:d}\|\big) - \psi(s_k)^\top w_k\Big]$;

**return** $w_K$

---

to either the objective function or the constraints, depending on whether the current policy violates any constraints:

- If the constraint $V_i^\pi$ $(i = 1, 2, \ldots, I)$ is violated, then the CRPO algorithm updates the violated constraint value function $V_i^\pi$.

- If all constraints are not violated, then the CRPO algorithm updates the objective value function $V_0^\pi$.

However, when constraints are near their boundaries, this method can lead to oscillations, making it difficult to track the performance of feasible policies and potentially resulting in unsafe policy outcomes when the model uncertainty presents. As the result, the algorithm cannot "remember" the highest objective value achieved by the feasible policy. To mitigate these limitations, our RRPO algorithm adopts a reformulated approach. Rather than following the standard CRPO routine, we leverage the constrained form of the original optimization problem to employ the CRPO algorithm as follows. We reformulate it into the following constrained maximization problem by introducing an auxiliary variable $d_0$:

$$\max_{d_0, \pi} \quad d_0$$
$$\text{s.t.} \quad V_0^\pi(\mu) \geq d_0,$$
$$V_i^\pi(\mu) \geq d_i, \text{ for } i = 1, 2, \ldots I.$$

At each iteration, the algorithm evaluates the robust value functions $V_i^{\pi_t}$ for all $i = 0, 1, \ldots, I$. Based on these evaluations, the algorithm proceeds in one of three categories:

1. **Threshold Updates:** If the current policy satisfies all constraints within a specified tolerance $\delta$ (that is, $V_i^{\pi_t} \geq d_i - \delta$ for $i = 0, 1, \ldots, I$), the algorithm updates the boundary threshold by setting $d_0 \leftarrow V_0^{\pi_t}(\mu)$.

2. **Constraint Rectification:** If any constraint is violated beyond the tolerance $\delta$ (that is, there exists $i = 1, 2, \ldots, I$, $V_i^{\pi_t}(\mu) > d_i - \delta$), the algorithm performs policy improvement steps to maximize the violated constraint, aiming to reduce constraint violation.

3. **Objective Rectification:** If the objective value $V_0^{\pi_t}(\mu)$ is less than the current best boundary threshold $d_0 - \delta$, the algorithm performs policy improvement steps to recover the objective value.

This procedure ensures that the policy maintains pursuing the feasibility while making progress towards optimizing the objective function. This procedure is summarized in Algorithm 1.

**Robust Policy Evaluation Subroutine** We consider the robust policy evaluation subroutine as a modular component decoupled from policy optimization. This procedure estimates the Q-function used in the natural policy gradient update. At each iteration, the agent interacts with the environment under the current policy $\pi_t$ to collect trajectories consisting of $(s, a, r, s')$ tuples. These samples are used to estimate the robust

Q-function $Q_i^{\pi_t}(s, a)$ for both the objective ($i = 0$) and each constraint ($i = 1, \ldots, I$). For value approximation, we adopt robust temporal-difference (TD) learning methods tailored to the chosen uncertainty model (e.g., $p$-norm or IPM uncertainty). Specifically, given a nominal transition model and an uncertainty set, we compute the worst-case expected value using closed-form solutions or dual formulations from prior work (e.g., Kumar et al. (2023); Zhou et al. (2024); Wang & Zou (2021)). These robust Q-estimates are then used to guide policy updates via a natural policy gradient step. The data collection procedure incorporated with this subroutine is further described in the Algorithm 1.

## 4.2 Handling Uncertainty

In our proposed algorithm design, we apply the robust natural policy gradient (Lemma 2, Zhou et al. (2024)) to maximize the value function. The update rule of maximizing $V_i^{\pi_t}(\mu)$ is given by

$$\pi_{t+1}(a|s) = \pi_t(a|s) \frac{\exp\left(\eta Q_i^{\pi_t}(s, a)/(1 - \gamma)\right)}{Z_t}, \tag{8}$$

where the normalization factor $Z_t$ is defined as $Z_t := \sum_{a \in \mathcal{A}} \pi_t(a|s) \exp\left(\eta Q_i^{\pi_t}(s, a)/(1 - \gamma)\right)$. When considering the softmax parametrization $\pi_\theta(a|s) := \frac{\exp(\theta(s, a))}{\sum_{a'} \exp(\theta(s, a'))}$, it is shown by Zhou et al. (2024) that this update rule is equivalent to

$$\theta_{t+1}(s, a) = \theta_t(s, a) + \eta Q_i^{\pi_t}(s, a), \tag{9}$$

where $\theta$ is taken over $\mathbb{R}^{\mathcal{S} \times \mathcal{A}}$. Throughout this paper, we will use the parametric and policy representations interchangeably. In updating the policy, accurate evaluation of $Q_i^\pi(s, a)$ is critical. To achieve this, we decouple the robust value function evaluation from the policy optimization step. This modular design allows us to integrate existing robust RL methods for value function approximation effectively.

Below, we highlight several promising approaches for approximating the robust value function:

- $p$-**norm uncertainty set** (Kumar et al., 2023): For each state-action pair $(s, a)$, define

$$\mathcal{U}_{(s,a)} = \left\{ u \in \mathbb{R}^{|\mathcal{S}|} \mid \langle u, \mathbf{1} \rangle = 0, \; \|u\|_p \leq \beta \right\}.$$

  Let $P_0$ be the nominal transition distribution. Then the corresponding uncertainty sets for transition probabilities are given by

$$\mathcal{P}_{(s,a)} := \left\{ P_0(\cdot \mid s, a) + u \mid u \in \mathcal{U}_{(s,a)} \right\}, \quad \text{and} \quad \mathcal{P} := \times_{(s,a) \in \mathcal{S} \times \mathcal{A}} \mathcal{P}_{(s,a)}. \tag{10}$$

  As shown in Proposition 2.3 of Kumar et al. (2023), the standard TD-learning algorithm can be applied, with adding a correction term, to compute the robust value function under this $p$-norm uncertainty model.

- **The integral probability metric (IPM) uncertainty set** (Zhou et al., 2024): Let $\mathcal{F} \subset \mathbb{R}^{|\mathcal{S}|}$ be a function class including the zero function. The IPM is defined as $d_{\mathcal{F}}(p, q) = \sup_{f \in \mathcal{F}}\{p^\top f - q^\top f\}$. The IPM uncertainty set is defined as

$$\mathcal{P}_{(s,a)} := \left\{ P_{s,a} \mid d_{\mathcal{F}}(P_{s,a}, P_0(\cdot|s, a)) \leq \beta \right\}, \quad \text{and} \quad \mathcal{P} := \times_{(s,a) \in \mathcal{S} \times \mathcal{A}} \mathcal{P}_{(s,a)}.$$

  Using the robust TD-learning algorithm (Algorithm 2, Zhou et al. (2024)), we can compute an approximate robust value function $\hat{V}_i^\pi(\mu)$. By leveraging the relationship between the robust value function and the robust Q-function (Proposition 2.2, Li et al. (2022)), along with the analytical worst-case formulation (Proposition 1, Zhou et al. (2024)), we can derive an approximate robust Q-function.

We acknowledge that other approaches also exist for approximating the robust Q-value function, such as Wang et al. (2023a); Sun et al. (2024); we omit these results due to the limited pages.

### 4.3 Global Convergence Guarantees

In this subsection, we establish the global convergence guarantee for RRPO under certain assumptions. Specifically, we assume: (1) The robust policy evaluation provides sufficiently accurate estimates. (2) Under the worst-case scenario, the policy still maintains sufficient exploration.

**Assumption 4.1** (Policy Evaluation Accuracy). The approximate robust value functions $\hat{Q}_i^\pi(s, a)$ satisfy $|\hat{Q}_i^\pi(s, a) - Q_i^\pi(s, a)| \leq \epsilon_{\text{approx}}$ for all $s \in \mathcal{S}$, $a \in \mathcal{A}$, and $i = 0, \dots, I$.

This assumption of sufficient accuracy in policy evaluation is mild and is widely adopted in the existing reinforcement learning literature (Wang et al., 2019; Cayci et al., 2022; Xu et al., 2021; Hong et al., 2023). As previously noted, this condition can be readily satisfied for specific uncertainty sets. We will discuss the value of $\epsilon_{\text{approx}}$ in the appendix.

**Assumption 4.2** (Worst-Case Exploration). For any policy $\pi$ and its worst-case transition $P$, there exists $p_{\min} > 0$ such that its state visitation probability satisfies $d_\mu^{\pi,P}(s) \geq p_{\min}$ for all $s \in \mathcal{S}$, where $d_\mu^{\pi,P}$ is the state visitation distribution starting from initial distribution $\mu$ under policy $\pi$ and transition $P$.

*Remark* 4.3. This assumption imposes a uniform lower bound for all policies $\pi$; another widely accepted assumption is made on the finite-horizon scenario (He et al., 2025), which adopt a complementary requirement that caps the relative re-weighting between nominal and adversarial dynamics. Exploring algorithms that retain our absolute-coverage guarantee while accommodating the ratio-based perspective of He et al. (2025) is an interesting avenue for future work.

The exploration assumption can hold under appropriate exploration mechanisms, especially with classical exploration techniques e.g. the initial state randomization; instead of a fixed initial state, we may use a uniform distribution over the state space, ensuring the state visitation probability is always lower bounded.

Moreover, this assumption *de facto* weakens some existing assumptions; for example, Theorem 2 (Chen & Huang, 2024) assumes there exists $p_{\min} > 0$ such that $\inf_{s,a,s'} P(s'|s, a) \geq p_{\min}$ for all uncertainty transition $P$. Assumption 2 (Zhou et al., 2024) assumes that the nominal transition is supported by all transitions in the given uncertainty set. If we are given the access to the robust policy evaluation oracle (Assumption 4.1), these widely used assumptions could be replaced with requiring the exploration on the worst-case transition instead of each transition in the uncertainty set.

**Assumption 4.4.** The diameter of the uncertainty set $c$ is given as

$$\sup_{P,P' \in C} \sup_{s,a} d_{\text{TV}}(P(\cdot|s, a), P'(\cdot|s, a)) \leq c.$$

This bounded-diameter condition restricts the variability of transition kernels within the uncertainty set. Intuitively, it ensures that all candidate models in $C$ are uniformly close to each other in terms of total variation distance, so the adversarial environment cannot deviate arbitrarily from the nominal dynamics. Such boundedness assumptions are standard in robust RL and are used by Ma et al. (2023), serving as a mild regularity condition to ensure well-posedness of the optimization problem.

Our main theoretical result is as follows. Here, we present a simplified version to highlight the most critical components, including the convergence rate and sample complexity. The detailed upper bound is provided in Appendix C.

**Theorem 4.5.** *Consider the NPG update rule Equation* (8) *with the learning rate* $\eta = \Theta(\frac{1}{\sqrt{T}})$. *Let the constraint violation tolerance* $\delta = \Theta(\frac{1}{\sqrt{T}}) + \mathcal{O}(c)$, *and the approximation error* $\epsilon_{approx} = \Theta(\frac{1}{\sqrt{T}})$. *Under these conditions, there exists an iteration* $T$ *such that the output policy* $\pi_{out}$ *satisfies*

$$\mathbb{E}[V^*(\mu) - V^{\pi_{out}}(\mu)] = \mathcal{O}(\frac{1}{\sqrt{T}}) + \mathcal{O}(c),$$

*where* $V^* := V^{\pi^*}$ *for the optimal feasible policy* $\pi^*$ *and*

$$\max_i \{d_i - V_i^{\pi_{out}}(\mu)\} \leq \delta.$$

*Remark* 4.6. Compared to Kitamura et al. (2025), which achieves a complexity of $\mathcal{O}(\epsilon^{-4})$ for exact convergence, our results establish a faster rate of $\mathcal{O}(\epsilon^{-2})$, up to an additional constant error determined by the uncertainty set diameter $c$. Specifically, our method converges to a neighborhood of the optimal policy, with the residual value error bounded by $c$. When $c$ is sufficiently small (e.g., when the uncertainty set is chosen to be small), our approach yields a strictly faster convergence rate up to an additive $\mathcal{O}(c)$ term. The full version of Theorem 4.5 and the detailed proof are provided in Appendix C.

**Discussion on the correction from the previous version of this work**   In our initial submission, we did not include the dependence on the uncertainty diameter $c$ in the upper bound. This oversight was identified through the OpenReview discussion (`https://openreview.net/forum?id=7l63xwAgAW`). Specifically, the problem arose because Lemma C.5 incorrectly upper bounded the advantage function when it takes negative values. In the current version, we have addressed this issue by introducing Assumption 4.4. Using the proposed RRPO (Algorithm 1), there is a constant gap linearly depending on the uncertainty diameter $c$, and cannot be closed for any $\epsilon$ less than this tolerance. Also, as indicated by Theorem 4 (Panaganti & Kalathil, 2022), if only learnt using the nominal model, one can show that the policy will incur another constant cost. Hence, the algorithm that solves CMDP (and thus, the CRPO) should have the same bound for this given constant.

### 4.4   Discussion

As shown in Theorem 4.5, to achieve $\epsilon$-accuracy to the optimal feasible policy $\pi^*$ up to a constant error term depending on the uncertainty diameter $c$, it takes at most $\mathcal{O}(\epsilon^{-2})$ iterations. Here, we use a specific uncertainty set to illustrate how the $\mathcal{O}(\epsilon^{-4})$ sample complexity is obtained. Assume we are considering the $(s, a)$-rectangular uncertainty set defined by the $p$-norm:

$$\mathcal{U}_{(s,a)} = \{u \in \mathbb{R}^{|\mathcal{S}|} \mid \langle u, \mathbf{1} \rangle = 0, \|u\|_p \leq \beta\}.$$

Let $P_0$ be the nominal distribution. Then

$$\mathcal{P}_{(s,a)} := \{P_0(\cdot|s,a) + u \mid u \in \mathcal{U}_{(s,a)}\}, \quad \text{and} \quad \mathcal{P} := \times_{(s,a) \in \mathcal{S} \times \mathcal{A}} \mathcal{P}_{(s,a)},$$

are the uncertainty set we consider. At each step $t$, we learn an $\epsilon$-accurate robust Q-function, which takes $\mathcal{O}(\epsilon^{-2})$ samples; this complexity is guaranteed by applying its Proposition 4.7, Kumar et al. (2023), to the standard TD-learning algorithm. Since Algorithm 1 requires $T = \mathcal{O}(\epsilon^{-2})$ iterations (to obtain an approximated optimal policy up to a $\mathcal{O}(c)$-level error) and the $\epsilon$-accurate policy evaluation requires $K = \mathcal{O}(\epsilon^{-2})$ samples, the total sample complexity is given by $T \cdot K = \mathcal{O}(\epsilon^{-4})$.

**Comparison with Kitamura et al. (2025)**   While Kitamura et al. (2025) also employs a constrained-form optimization framework (called the *epigraph* formulation), our approach is fundamentally different. (1) Their method updates the maximum constraint violation, while we adopt the original CRPO update rule with tracking the auxiliary parameter. (2) Their algorithm requires an additional binary search step, while our method preserves the simplicity of the CRPO framework without such overhead.

## 5   Numerical Examples

To better illustrate the impact of model uncertainty on the algorithm performance, especially on the worst-case feasibility, we conducted experiments comparing the proposed RRPO and the CRPO (Xu et al., 2021). For all experiments, we used a discount factor $\gamma = 0.99$ and the 2-norm uncertainty set defined by Equation (10).

### 5.1   The FrozenLake-Like Gridworld

First, we consider a specific $4 \times 6$ FrozenLake-like gridworld environment: The agent starts from the left-top corner $\mathsf{pos}_{\text{start}} = [1, 1]$ and can make four actions,

$$\mathcal{A} = \{\text{UP}, \text{DOWN}, \text{LEFT}, \text{RIGHT}\},$$

to move to the target point $\mathsf{pos}_{\text{target}} = [2, 5]$.

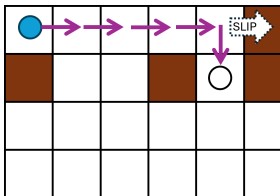 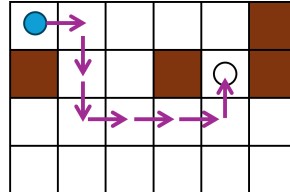

Figure 2: Illustration of two paths to the target in the grid-world environment. The shortest path (Left) prioritizes efficiency but risks violating constraints in slippery conditions, whereas the longer path (Right) always ensures safety in the worst-case scenario.

We define two reward functions. The main reward function $r_0$ is defined as

$$r_0(s, a, s') = \begin{cases} +1 & \text{if } s' \text{ is the target} \\ -1 & \text{if } s' \text{ is a brown block} \\ -0.1 & \text{otherwise} \end{cases}.$$

It gives $+1$ for reaching the target, $-1$ for landing on a brown block, and $-0.1$ otherwise. The constraint reward function $r_1$ is defined as:

$$r_1(s, a, s') = \begin{cases} -1 & \text{if } s' \text{ is out of the boundary} \\ -1 & \text{if } s' \text{ is a brown block} \\ 0 & \text{otherwise} \end{cases}.$$

It assigns $-1$ for stepping out of the boundary or onto a brown block, and $0$ otherwise, leading to a cost function $c(s, a) := -\mathbb{E}[r_1(s, a, s')]$. We require $-V_1^{\pi}(\mu) < 0.2$, ensuring the agent avoids hitting brown blocks or moves out of the boundary. The training environment is deterministic, where each action leads to the intended movement with probability one unless the agent hits a boundary or a brown block. When this happens, the agent's position is not changed (if it hits the boundary) or is reset to the starting position (if it hits the brown block). The test environment introduces a "slippery" dynamic, where every move has a probability $p$ of resulting in an unintended slip. This slippery setting mimics conditions that may not have been foreseen during training, effectively representing a worst-case scenario. Under this setting, the obstacles construct two distinct paths routing to the target, which is illustrated in Figure 2.

We apply our proposed RRPO to solve this constrained robust RL problem, comparing it with the baseline CRPO method. As shown in Figure 3a, our method successfully learns the safer path, while the non-robust algorithm converges to the shortest path.

## 5.2 Mountain Car

We also consider the classical Mountain Car environment from Gymnasium (Towers et al., 2024) to test the performance of the proposed RRPO in the classical control problem. We use its default reward function $r_0$, which penalizes $-1.0$ each step and rewards $0$ if the agent reaches the goal; that is,

$$r_0(s, a, s') = \begin{cases} 0 & \text{if the agent reaches the goal,} \\ -1 & \text{otherwise.} \end{cases}$$

To emphasize safety, we add a constraint reward function $r_1(s, a, s')$ defined as

$$r_1(s, a, s') = \begin{cases} -1 & \text{if the car's speed exceeds } 0.06, \\ 0 & \text{otherwise.} \end{cases}$$

It returns $-1.0$ whenever the car's speed exceeds $0.06$ and returns $0$ otherwise, which encourages the agent to maintain a safe speed throughout its run. We also consider its cost description as $c(s, a) := -\mathbb{E}_{s'}[r_1(s, a, s')]$.

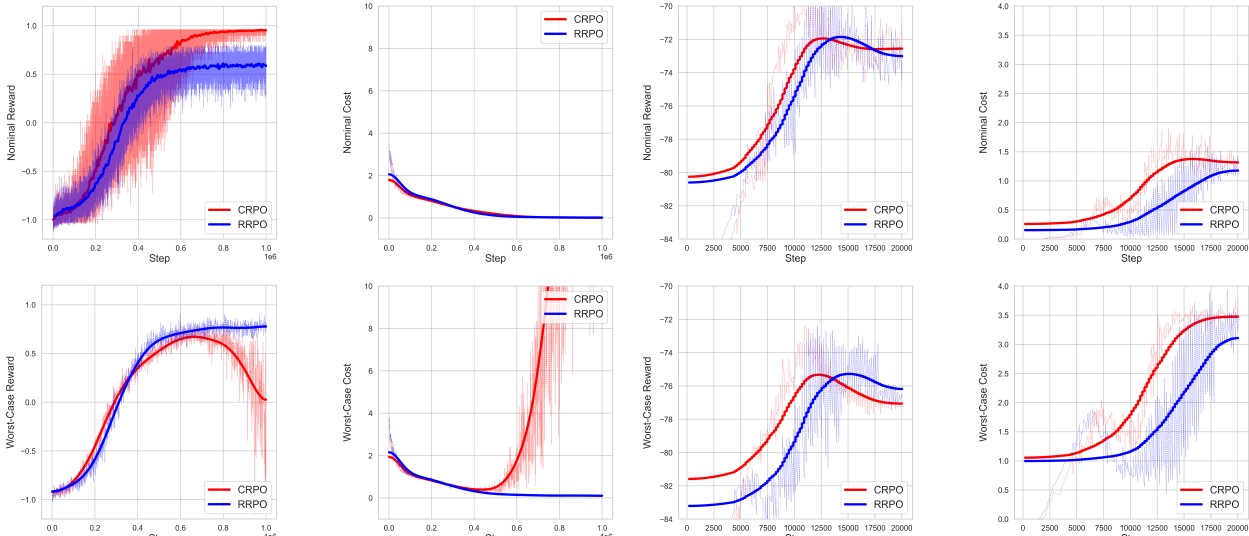

(a) Illustration of two paths to the target in the grid-world environment. The shortest path (Left) prioritizes efficiency but risks violating constraints in slippery conditions, whereas the longer path (Right) always ensures safety in the worst-case scenario.

(b) Reward and cost comparison across nominal and worst-case transitions for the mountain car environment. In the nominal environment, both algorithms learn the desired strategies to reach the goal; however, in the worst-case scenario, the RRPO algorithm can learn more robust strategy to avoid exceeding the speed constraint.

Figure 3: Reward and cost trajectories for CRPO and RRPO under nominal and worst-case transitions in Grid World (Figure 3a) and Mountain Car (Figure 3b) environments.

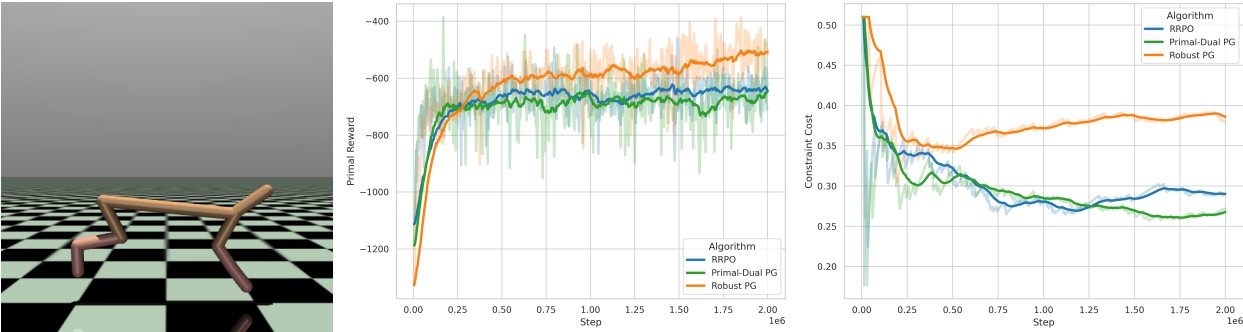

Figure 4: Comparison among our proposed RRPO algorithm and other robust RL baselines in the robust HalfCheetah environment (shown in the left panel). The standard robust policy gradient algorithm achieves the highest score but violates the energy constraint as shown in the right panel. Within the given constraint, RRPO outperforms the primal-dual approach as shown in the middle panel.

For the environment uncertainty, we perturb the "gravity" parameter of the Mountain Car environment. In the worst-case scenario, the gravity is increased from the nominal value 0.0025 to 0.003. The experiment results are shown in Figure 3b; the proposed RRPO method receives much less cost in the worst-case environment.

## 6 Extended Experiments: Robust Control

Additionally, we compare the performance of the RRPO algorithm with other robust RL baselines, including the robust policy gradient and the robust primal-dual policy gradient methods, on the HalfCheetah-v4 environment from Robust Gymnasium (Gu et al., 2025). We also conduct an ablation study to evaluate the

impact of our modification to the original robust CRPO algorithm, demonstrating that it effectively reduces oscillations near the constraint boundary. Although the Safety MuJoCo environment (Gu et al., 2024a), also available in Robust Gymnasium, offers benchmarks with hard safety constraints, its formulation requires additional adaptation to fit our setting. Therefore, we employ a custom energy constraint that penalizes energy consumption at each step. We include the experiment details and the hyper-parameter setting in Appendix D.4.

## 6.1 Comparison with Other Robust RL Algorithms

We conducted a comparative evaluation of three robust RL algorithms on the HalfCheetah-v4 environment from Robust Gymnasium (Gu et al., 2025): (i) Robust Policy Gradient. (ii) Robust Primal-Dual Policy Gradient. (iii) Our proposed RRPO algorithm.

The agent learns to maximize forward locomotion speed while respecting energy consumption constraints. The reward function follows the standard HalfCheetah formulation:

$$r_0(s, a) = \text{forward\_velocity} - 0.1 \|a\|^2,$$

encouraging rapid forward movement while penalizing excessive control effort. Crucially, we implement a separate energy constraint defined as the squared $L_2$ norm of the action vector

$$r_1(s, a) = -\|a\|^2$$

with a threshold of $-0.25$ (note that this threshold is applied to the value function instead of the current step cost; it does not apply to the constraint cost shown in Figure 3). We include the detailed hyper-parameter setting in Appendix D.4.

## 6.2 Ablation: Robust CRPO Algorithm

In the ablation experiment, we compare our RRPO algorithm with the **Robust version of CRPO algorithm** (that is, we replace the value function used in the CRPO algorithm with the robust value function). The only difference presented here is the rectification mechanism used in Algorithm 1 where the RRPO algorithm tracks the best feasible policy as the $\pi_{\text{out}}$ in the threshold update step. We follow the same setting as described in Appendix D.4. To compare both methods, we evaluate the variance of the primal reward of the output policy. In the RRPO algorithm, this variance primarily stems from inherent randomness in the policy and environment. In CRPO, however, an additional source of variance arises from the random sampling of the output policy. As a result, CRPO exhibits substantially higher variance than RRPO.

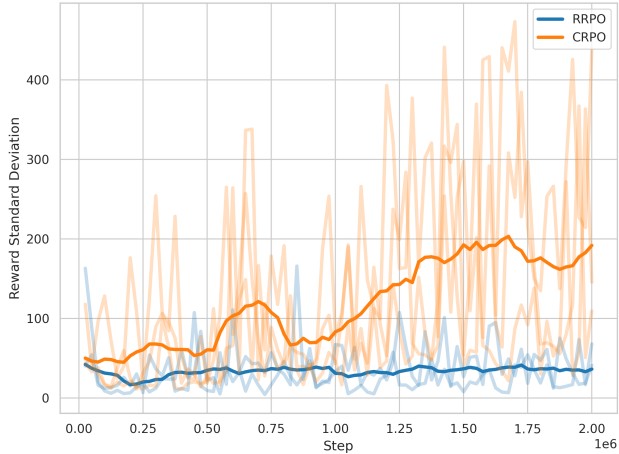

Figure 5: Comparison of the variance of the primal reward $r_0$ of the output policy from CRPO (Xu et al., 2021) and the RRPO (Algorithm 1). The CRPO method exhibits substantially higher variance than RRPO.

## 7   Conclusion

In this paper, we investigated robust constrained RL problems and demonstrated that strong duality generally **DOES NOT** hold in this setting, thereby limiting the effectiveness of some methods which rely on the strong duality. To address this challenge, we introduced RRPO, a primal-only algorithm that directly optimizes the policy while rectifying constraint violations without relying on dual formulations. Our theoretical analysis, under mild boundedness and exploration assumptions, provided convergence guarantees for RRPO, ensuring that it converges to an approximately optimal policy up to a $\mathcal{O}(c)$-level value function error that satisfies the constraints within a specified tolerance under worst-case scenarios. Empirical results validate the effectiveness of our approach. We believe our work opens new avenues for exploring and designing non-primal-dual approaches to solve robust constrained RL problems, and potentially leads to an interesting direction to identify when the strong duality of robust constrained RL holds.

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

## A  Further Discussions on Related Work

In this section, we include further discussions on existing literature include other closely related areas and clarification of existing results on the strong duality of robust constrained RL problems.

### A.1  Other Related Work

In this section, we further explore the two closely related areas: robust RL and constrained RL.

**Robust RL**    Robust reinforcement learning (RL) aims to develop policies that perform well under the worst-case transitions. Early works on robust RL primarily focused on model-based approaches, where the uncertainty set of transition probabilities is known or can be estimated, and robust policies are computed using robust dynamic programming techniques (Bagnell et al., 2001; Iyengar, 2005; Nilim & Ghaoui, 2005; Satia & Lave Jr., 1973; Wiesemann et al., 2013; Lim & Xu, 2013). These methods consider worst-case scenarios over the uncertainty set to ensure robustness. In the model-free setting, robust RL algorithms have been proposed that do not require explicit knowledge of the uncertainty set but instead utilize samples to estimate robust value functions and policies (Roy et al., 2017; Wang & Zou, 2021; Panaganti & Kalathil, 2022). These methods often involve solving a robust optimization problem over the estimated uncertainties. Recent theoretical advancements overcome the issues of directly solving the worst-case transitions. Wang & Zou (2022) considers the $R$-contamination model to obtain the unbiased estimator for the policy gradient method. Zhou et al. (2024) applies the double sampling method or the structure of the IPM uncertainty structure to obtain the unbiased estimation involving the worst-case transition probability. And Kumar et al. (2023) provides the analytical solution for the $p$-norm uncertainty set. These advancements allow us to directly obtain the robust policy gradient or the robut value function without estimating the worst-case transition probability.

**Constrained RL**    Constrained reinforcement learning extends the standard RL framework by incorporating constraints into the agent's decision-making process, aiming to optimize performance while satisfying certain safety, resource, or risk constraints (Altman, 1999). A widely used approach for solving constrained RL problems is the primal-dual method (Paternain et al., 2019; Tessler et al., 2019; Liang et al., 2018; Stooke et al., 2020), which leverages the strong duality property of constrained RL (Altman, 1999) to formulate a Lagrangian that combines the objective function with the weighted constraints. These methods iteratively update the policy and the Lagrange multipliers, and convergence guarantees have been established under certain conditions (Ding et al., 2020; Liu et al., 2021). However, these methods rely on the assumption of strong duality, which may not hold in more complex settings. Alternative methods, known as primal methods, enforce constraints directly by projecting policies onto the feasible set or using safe policy improvement techniques (Achiam et al., 2017; Chow et al., 2018b; Dalal et al., 2018b; Xu et al., 2021; Yang & Zhang, 2020). These methods aim to ensure constraint satisfaction without relying on dual variables.

### A.2    Strong Duality in Robust Constrained RL Problems

In this section, we provide more detailed discussions in the existing literature discussing the strong duality in robust constrained RL problems.

In the existing literature, Ghosh (2024) provide an intuitive explanation for why existing primal-dual methods for non-robust constrained RL problems could fail in robust case: In the standard routine of showing the strong duality (Paternain et al., 2019; Altman, 1999), the state visitation distribution $d^{\pi,P}$ is convex in the policy $\pi$; that is, there always exists a policy $\pi'$ such that $(1-\alpha)d^{\pi,P} + \alpha d^{\pi,P} = d^{\pi',P}$. However, this relation obviously does not hold, which makes the strong duality of robust constrained RL problems unclear. Our counterexample offers a theoretical justification of this conjecture by providing a concrete example where the duality gap is strictly positive.

Additionally, Zhang et al. (2024) has proved the strong duality for robust constrained RL problems by employing the "randomization trick" that modifies the optimization problem's policy space. However, their results do not apply to our setting. Specifically, it doesn't consider the space of all random policies; instead, it only considers the distribution of deterministic policies. The choice of deterministic policy is made at the beginning of each round. This approach redefines the robust constrained RL problem to ensure strong duality, differing from the classical definition used in constrained RL (Altman, 1999; Paternain et al., 2019). Our work, instead, aims to align with this classical definition, highlighting that without such extensions, strong duality may not hold.

As the result, existing literature has not addressed the critical question in the robust constrained RL problems, which is what we aim to solve in this paper.

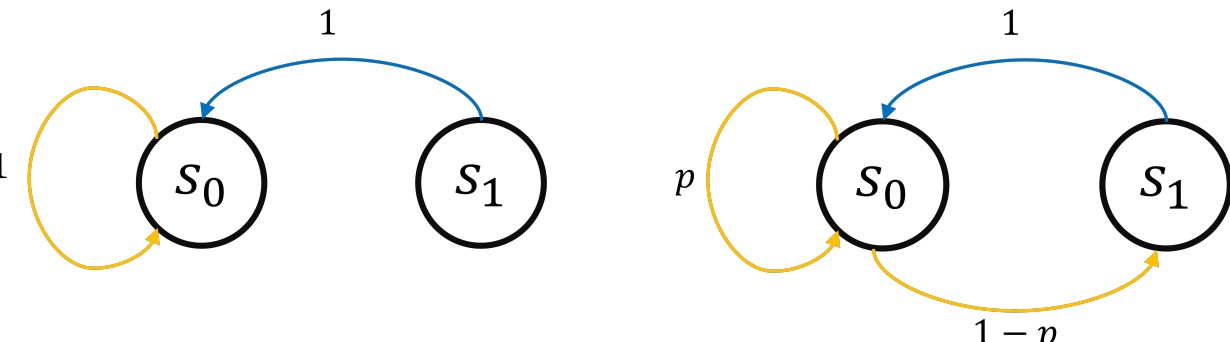

(a) Transition when taking the action $a_0$.      (b) Transition when taking the action $a_1$.

Figure 6: The transition diagram of the MDP considered in Theorem 3.1. At state $s_1$, the agent always moves to state $s_0$ with probability 1, regardless of the action taken. At state $s_0$, the agent has a probability $p$ of staying in the current state when taking action $a_1$, and a probability of 1 of staying in state $s_0$ when taking action $a_0$. The uncertainty only occurs in the transition probability $p$; we let it vary from $[\underline{p}, \overline{p}]$.

## B    Counterexample: Robust Constrained RL with Non-Zero Duality Gap

*Proof.* We divide the proof into three parts: (1) The construction of counterexample constrained robust MDP. (2) The evaluation of Lagrangian function. (3) The evaluation of the duality gap.

1. **Construction of the constrained robust MDP:**

   We consider the constrained robust MDP described in Figure 6 (which is the same as Figure 1). The nominal transition probability is explicitly defined as follows:

$$P(s_0 \mid a_0, s_0) = 1,$$
$$P(s_1 \mid a_0, s_0) = 0,$$
$$P(s_0 \mid a_1, s_0) = p,$$
$$P(s_1 \mid a_1, s_0) = 1 - p,$$
$$P(s_0 \mid a_0, s_1) = 1,$$
$$P(s_1 \mid a_0, s_1) = 0,$$
$$P(s_0 \mid a_1, s_1) = 1,$$
$$P(s_1 \mid a_1, s_1) = 0.$$

   Then we obtain the state transition probability induced by the policy $\pi$:

$$P^\pi = \begin{bmatrix} \pi_0 + \pi_1 p & 1 \\ \pi_1(1-p) & 0 \end{bmatrix},$$

   where $\pi_0 := \pi(a_0|s_0)$ and $\pi_1 := \pi(a_1|s_0)$. The action at the state $s_1$ doesn't make any differences. The $(i,j)$-th entry of $P^\pi$ represents the probability of moving from $s_j$ to $s_i$ by following the policy $\pi$. We fix the initial state to $s_0$. Its corresponding distribution is given by $\mu_0 = \begin{bmatrix} 1 \\ 0 \end{bmatrix}$. The reward for the objective value function is $r_0 = \begin{bmatrix} 1 \\ 0 \end{bmatrix}$. The reward for the constraint is $r_1 = \begin{bmatrix} 0 \\ 1 \end{bmatrix}$. Here, we only consider a single constraint.

   Lastly, we consider the following $(s,a)$-uncertainty set defined by the $L_\infty$ distance:

$$\mathcal{V}_{s,a} := \{0\}, \text{ for } (s,a) \neq (s_0, a_1)$$

$$\mathcal{V}_{s_0,a_1} := \{v \in R^{|S|} \mid \langle v, 1_{|S|} \rangle = 0, \|v\|_\infty \le \beta\},$$
$$\mathcal{U}_{s,a} := \mathcal{V}_{s,a} + P.$$

Given this uncertainty set, the value of $p$ varies from $p - \delta$ to $p + \delta$. Here, we further assume that $p$ is strictly less than 1, $\delta < p$, and $p + \delta < 1$. We denote $\overline{p} := p + \delta$ and $\underline{p} := p - \delta$.

2. **Evaluate the Lagrangian function:**

Now we evaluate the discounted visitation measure of the policy $\pi$. Define the discounted visitation measure as $d := (I - \gamma P^\pi)^{-1} \mu_0$. Because $I - \gamma P^\pi = \begin{bmatrix} 1 - \gamma(\pi_0 + \pi_1 p) & -\gamma \\ -\gamma \pi_1 (1 - p) & 1 \end{bmatrix}$, where $\mu_0$ is the initial state distribution. Then we obtain

$$d = \frac{1}{|I - \gamma P^\pi|} \begin{bmatrix} 1 & \gamma \\ \gamma \pi_1 (1 - p) & 1 - \gamma + \gamma \pi_1 (1 - p) \end{bmatrix} \begin{bmatrix} 1 \\ 0 \end{bmatrix} = \frac{1}{|I - \gamma P^\pi|} \begin{bmatrix} 1 \\ \gamma \pi_1 (1 - p) \end{bmatrix}.$$

Here, $|\cdot|$ represent the determinant. And we choose $\gamma$ to ensure $\|\gamma P^\pi\|_2 < 1$.

Given the discounted visitation measure, we immediately obtain the non-robust value function of each reward by using $V^\pi(\mu_0) = r^\top d$:

$$V_0^\pi(\mu_0) = \frac{1}{1 - \gamma + \pi_1 (1 - p)(\gamma - \gamma^2)},$$
$$V_1^\pi(\mu_0) = \frac{\gamma \pi_1 (1 - p)}{1 - \gamma + \pi_1 (1 - p)(\gamma - \gamma^2)}.$$

From our definition of the uncertainty set, we have $p \in [\underline{p}, \overline{p}]$. Then the robust value function of each reward is:

$$\widetilde{V}_0^\pi(\mu_0) = \min_p V_0^\pi(\mu_0) = \frac{1}{1 - \gamma + \pi_1 (1 - \underline{p})(\gamma - \gamma^2)},$$
$$\widetilde{V}_1^\pi(\mu_0) = \min_p V_1^\pi(\mu_0) = \frac{\gamma \pi_1 (1 - \overline{p})}{1 - \gamma + \pi_1 (1 - \overline{p})(\gamma - \gamma^2)}.$$

Therefore, the Lagrangian function is given by

$$\mathcal{L}(\pi, \lambda) = \widetilde{V}_0^\pi - \lambda(\rho - \widetilde{V}_1^\pi)$$
$$= \frac{1}{1 - \gamma + \pi_1 (1 - \underline{p})(\gamma - \gamma^2)} - \lambda \left( \rho - \frac{\gamma \pi_1 (1 - \overline{p})}{1 - \gamma + \pi_1 (1 - \overline{p})(\gamma - \gamma^2)} \right).$$

3. **Evaluate the duality gap:**

It suffices to evaluate both $\min_\lambda \max_\pi \mathcal{L}(\pi, \lambda)$ and $\max_\pi \min_\lambda \mathcal{L}(\pi, \lambda)$.

- Solve $\min_\lambda \max_\pi \mathcal{L}(\pi, \lambda)$:
  We will solve $\frac{\partial \mathcal{L}}{\partial \pi} \ge 0$ to find monotone intervals of the function $\mathcal{L}(\cdot, \lambda) : \pi \mapsto \mathcal{L}(\pi, \lambda)$. Then we will obtain when $\mathcal{L}(\cdot, \lambda) : \pi \mapsto \mathcal{L}(\pi, \lambda)$ achieves its maxima.

$$\frac{\partial \mathcal{L}}{\partial \pi} = \frac{-(1 - \underline{p})(\gamma - \gamma^2)}{\left[ 1 - \gamma + \pi_1 (1 - \underline{p})(\gamma - \gamma^2) \right]^2}$$
$$+ \lambda \frac{\gamma(1 - \overline{p}) \left[ 1 - \gamma + \pi_1 (1 - \overline{p})(\gamma - \gamma^2) \right] - \gamma \pi_1 (1 - \overline{p}) \left[ (1 - \overline{p})(\gamma - \gamma^2) \right]}{\left[ 1 - \gamma + \pi_1 (1 - \overline{p})(\gamma - \gamma^2) \right]^2}$$
$$= \frac{-(1 - \underline{p})(\gamma - \gamma^2)}{\left[ 1 - \gamma + \pi_1 (1 - \underline{p})(\gamma - \gamma^2) \right]^2} + \lambda \frac{(1 - \overline{p})(\gamma - \gamma^2)}{\left[ 1 - \gamma + \pi_1 (1 - \overline{p})(\gamma - \gamma^2) \right]^2}.$$

Let $\frac{\partial \mathcal{L}}{\partial \pi} \ge 0$. We obtain

$$\lambda \frac{1 - \overline{p}}{\left[ 1 + \pi_1 (1 - \overline{p})\gamma \right]^2} \ge \frac{1 - \underline{p}}{\left[ 1 + \pi_1 (1 - \underline{p})\gamma \right]^2}$$

$$\implies \quad \lambda \frac{1-\overline{p}}{1-\underline{p}} \left[1 + \pi_1(1-\underline{p})\gamma\right]^2 \geq \left[1 + \pi_1(1-\overline{p})\gamma\right]^2$$

$$\implies \quad \sqrt{\lambda \frac{1-\overline{p}}{1-\underline{p}}} \left[1 + \pi_1(1-\underline{p})\gamma\right] \geq \left[1 + \pi_1(1-\overline{p})\gamma\right].$$

Then we obtain

$$\sqrt{\lambda \frac{1-\overline{p}}{1-\underline{p}}} - 1 \geq \pi_1 \gamma \left[(1-\overline{p}) - (1-\underline{p})\sqrt{\lambda \frac{1-\overline{p}}{1-\underline{p}}}\right].$$

It is easy to notice that when $\lambda \leq \frac{1-\overline{p}}{1-\underline{p}}$, the coefficient of $\pi_1$

$$\left[(1-\overline{p}) - (1-\underline{p})\sqrt{\lambda \frac{1-\overline{p}}{1-\underline{p}}}\right] \geq 0.$$

When $\lambda \geq \frac{1-\overline{p}}{1-\underline{p}}$, the coefficient of $\pi_1$

$$\left[(1-\overline{p}) - (1-\underline{p})\sqrt{\lambda \frac{1-\overline{p}}{1-\underline{p}}}\right] \leq 0.$$

We separately consider each case to solve the monotone intervals.

∘ Case 1: $\lambda \leq \frac{1-\overline{p}}{1-\underline{p}}$. In this case, $(1-\overline{p}) - (1-\underline{p})\sqrt{\lambda \frac{1-\overline{p}}{1-\underline{p}}} \geq 0$. It solves:

$$\pi_1 \leq \frac{\left[\sqrt{\lambda \frac{1-\overline{p}}{1-\underline{p}}} - 1\right]}{\gamma \left[(1-\overline{p}) - (1-\underline{p})\sqrt{\lambda \frac{1-\overline{p}}{1-\underline{p}}}\right]}.$$

We further consider if this upper bound is positive or negative.

▷ Case 1.1: $\lambda \geq \frac{1-\underline{p}}{1-\overline{p}}$. In this case, $\left[\sqrt{\lambda \frac{1-\overline{p}}{1-\underline{p}}} - 1\right] \geq 0$. It violates $\lambda \leq \frac{1-\overline{p}}{1-\underline{p}}$.

▷ Case 1.2: $\lambda \leq \frac{1-\underline{p}}{1-\overline{p}}$. In this case, $\left[\sqrt{\lambda \frac{1-\overline{p}}{1-\underline{p}}} - 1\right] \leq 0$.

Therefore, in the Case 1 ($\lambda \leq \frac{1-\overline{p}}{1-\underline{p}}$), we always have $\pi_1 \leq 0$. It indicates that $\mathcal{L}(\pi, \lambda)$ is decreasing in $\pi_1$ when $\pi_1 \in [0,1]$. The maximum is achieved when setting $\pi_1 = 0$. That is,

$$\max_\pi \mathcal{L}(\pi, \lambda) = \mathcal{L}(0, \lambda) = \frac{1}{1-\gamma} - \lambda\rho,$$

where $0 \leq \lambda \leq \frac{1-\overline{p}}{1-\underline{p}}$.

∘ Case 2: $\lambda \geq \frac{1-\overline{p}}{1-\underline{p}}$. In this case, $(1-\overline{p}) - (1-\underline{p})\sqrt{\lambda \frac{1-\overline{p}}{1-\underline{p}}} \leq 0$. It solves:

$$\pi_1 \geq \frac{\left[\sqrt{\lambda \frac{1-\overline{p}}{1-\underline{p}}} - 1\right]}{\gamma \left[(1-\overline{p}) - (1-\underline{p})\sqrt{\lambda \frac{1-\overline{p}}{1-\underline{p}}}\right]}.$$

Again, we further consider if this lower bound is positive or negative:

▷ Case 2.1: $\lambda \geq \frac{1-\underline{p}}{1-\overline{p}}$. In this case, $\left[\sqrt{\lambda \frac{1-\overline{p}}{1-\underline{p}}} - 1\right] \geq 0$. It indicates that $\mathcal{L}(\pi, \lambda)$ is increasing in $\pi_1$ when $\pi_1 \in [0,1]$. The maximum is achieved at $x_1 = 1$.

▷ Case 2.2: $\frac{1-\overline{p}}{1-\underline{p}} \leq \lambda \leq \frac{1-\underline{p}}{1-\overline{p}}$. In this case, $\left[\sqrt{\lambda \frac{1-\overline{p}}{1-\underline{p}}} - 1\right] \leq 0$. It indicates that $\mathcal{L}(\pi, \lambda)$ is decreasing then increasing within $\pi_1 \in [0,1]$. The maximum is either achieved at $x_1 = 1$ or $x_1 = 0$. We need to decide which one is larger: When $x_1 = 1$, we have

$$\mathcal{L}(1, \lambda) = \frac{1}{1-\gamma + (1-\underline{p})(\gamma - \gamma^2)} - \lambda\left(\rho - \frac{\gamma(1-\overline{p})}{1-\gamma + (1-\overline{p})(\gamma - \gamma^2)}\right)$$

or when $x_1 = 0$,

$$\mathcal{L}(0, \lambda) = \frac{1}{1 - \gamma} - \lambda\rho.$$

By letting $\mathcal{L}(1, \lambda) \geq \mathcal{L}(0, \lambda)$, we solve the boundary is

$$\hat{\lambda} = \frac{1 - \underline{p}}{1 - \overline{p}} \frac{1 + (1 - \overline{p})\gamma}{1 + (1 - \underline{p})\gamma}.$$

It means if $\lambda \geq \hat{\lambda}$, then $\mathcal{L}(1, \lambda) \geq \mathcal{L}(0, \lambda)$. If $\lambda \leq \hat{\lambda}$, then $\mathcal{L}(1, \lambda) \leq \mathcal{L}(0, \lambda)$. Combining both Case 2.1 and Case 2.2, we obtain

$$\max_{\pi} \mathcal{L}(\pi, \lambda) = \begin{cases} \mathcal{L}(1, \lambda) & \lambda \geq \hat{\lambda} \\ \mathcal{L}(0, \lambda) & \hat{\lambda} \geq \lambda \geq \frac{1 - \overline{p}}{1 - \underline{p}} \end{cases},$$

where $\hat{\lambda} = \frac{1 - \underline{p}}{1 - \overline{p}} \frac{1 + (1 - \overline{p})\gamma}{1 + (1 - \underline{p})\gamma}$.

Now, combining both Case 1 and Case 2, we have

$$\max_{\pi} \mathcal{L}(\pi, \lambda) = \begin{cases} \mathcal{L}(1, \lambda) & \lambda \geq \hat{\lambda} \\ \mathcal{L}(0, \lambda) & \hat{\lambda} \geq \lambda \geq 0 \end{cases},$$

where $\hat{\lambda} = \frac{1 - \underline{p}}{1 - \overline{p}} \frac{1 + (1 - \overline{p})\gamma}{1 + (1 - \underline{p})\gamma}$. When $\lambda = \hat{\lambda}$, the function $\max_{\pi} \mathcal{L}(\pi, \cdot) : \lambda \mapsto \max_{\pi} \mathcal{L}(\pi, \lambda)$ achieves its minimum. That is,

$$\min_{\lambda} \max_{\pi} \mathcal{L}(\pi, \lambda) = \frac{1}{1 - \gamma} - \frac{1 - \underline{p}}{1 - \overline{p}} \frac{1 + (1 - \overline{p})\gamma}{1 + (1 - \underline{p})\gamma}\rho.$$

- Solve $\max_{\pi} \min_{\lambda} \mathcal{L}(\pi, \lambda)$:
  We let the constraint be satisfied; that is

$$\rho - \frac{\gamma\pi_1(1 - \overline{p})}{1 - \gamma + \pi_1(1 - \overline{p})(\gamma - \gamma^2)} \leq 0.$$

Otherwise, simply letting $\lambda = +\infty$ will lead to $-\infty$ function value. It solves

$$\rho(1 - \gamma) \leq [1 - \rho(1 - \gamma)]\gamma(1 - \overline{p})\pi_1.$$

Since $\rho$ must be less than $\frac{1}{1 - \gamma}$ (so, $1 - (1 - \gamma)\rho \geq 0$), we have

$$\pi_1 \geq \frac{\rho(1 - \gamma)}{[1 - \rho(1 - \gamma)]\gamma(1 - \overline{p})}.$$

Therefore, the maximum is achieved at $\pi_1 = \frac{\rho(1 - \gamma)}{[1 - \rho(1 - \gamma)]\gamma(1 - \overline{p})}$:

$$\max_{\pi} \min_{\lambda} \mathcal{L}(\pi, \lambda) = \max_{\pi} \widetilde{V}_0^{\pi}$$

$$= \frac{1}{1 - \gamma + \frac{\rho(1 - \gamma)^2}{1 - \rho(1 - \gamma)} \frac{1 - \underline{p}}{1 - \overline{p}}}$$

$$= \frac{1}{1 - \gamma} - \frac{\rho\frac{1 - \underline{p}}{1 - \overline{p}}}{1 - \rho(1 - \gamma) + \rho(1 - \gamma)\frac{1 - \underline{p}}{1 - \overline{p}}}.$$

Therefore, the duality gap is given by

$$\mathscr{D}(\mathcal{L}) = \max_{\pi} \min_{\lambda} \mathcal{L}(\pi, \lambda) - \min_{\lambda} \max_{\pi} \mathcal{L}(\pi, \lambda)$$

$$= \frac{1 - \underline{p}}{1 - \overline{p}} \frac{1 + (1 - \overline{p})\gamma}{1 + (1 - \underline{p})\gamma}\rho - \frac{\rho\frac{1 - \underline{p}}{1 - \overline{p}}}{1 - \rho(1 - \gamma) + \rho(1 - \gamma)\frac{1 - \underline{p}}{1 - \overline{p}}}.$$

When $\bar{p} = \underline{p}$, the duality gap turns to be exactly 0. It is because the constrained non-robust RL problem has zero duality gap. However, when we set $p = \gamma = 0.5$, $\bar{p} = 0.75$, $\underline{p} = 0.25$, and $\rho = 1$. We have the non-zero duality gap

$$\mathscr{D}(\mathcal{L}) = \frac{21}{22}.$$

$\square$

## C  Proof of Theorem 4.5

In this section, we provide the detailed proof of Theorem 4.5.

### C.1  Assumptions

In this subsection, we recap the assumptions used in this proof. We additionally restrict all rewards to $[0, 1]$; however, this restriction is not crucial. It only affects the constant upper bound of robust value (or Q) functions $V^\pi$ and $Q^\pi$. We can relax this assumption to a general $[-r_{\max}, r_{\max}]$ with changing the upper and lower bound of these value functions to be $\frac{r_{\max}}{1-\gamma}$.

**Assumption C.1** (Policy Evaluation Accuracy). The approximate robust value functions $\hat{Q}_i^\pi(s, a)$ satisfy $|\hat{Q}_i^\pi(s, a) - Q_i^\pi(s, a)| \leq \epsilon_{\text{approx}}$ for all $s \in \mathcal{S}$, $a \in \mathcal{A}$, and $i = 0, \dots, I$.

**Assumption C.2** (Worst-Case Exploration). For any policy $\pi$ and its worst-case transition $P$, there exists a positive constant $p_{\min} > 0$ such that its state visitation probability satisfies $d_\mu^{\pi,P}(s) \geq p_{\min}$ for all $s \in \mathcal{S}$, where $d_\mu^{\pi,P}$ is the state visitation distribution starting from initial distribution $\mu$ under policy $\pi$ and transition $P$.

**Assumption C.3** (Bounded Rewards). For all rewards $r_i : \mathcal{S} \times \mathcal{A} \to \mathbb{R}$, it satisfies

$$0 \leq r_i(s, a) \leq 1$$

for all $(s, a) \in \mathcal{S} \times \mathcal{A}$.

**Assumption C.4.** The diameter of the uncertainty set $C$ is given as

$$\sup_{P, P' \in C} \sup_{s,a} d_{\text{TV}}(P(\cdot|s, a), P'(\cdot|s, a)) \leq c.$$

### C.2  Supporting Lemmas

We summarize all required lemmas in this subsection. These lemmas will be used to prove the main result.

The following performance difference lemma is originally developed by Zhou et al. (2024) for bounding the value function difference of two policies $\pi'$ and $\pi$. Here we apply Assumption 4.2 to turn the upper and lower bound to transition this inequality to the desired distribution needed in our convergence analysis.

**Lemma C.5** (Robust performance difference lemma). *Let $\pi, \pi'$ be two policies and $P, P'$ be their worst-case transition kernels. Suppose that $\mu$ is the initial distribution over the state space $S$. Then*

$$\frac{1}{1-\gamma} \mathbb{E}_{s \sim d_\mu^{\pi',P'}} \mathbb{E}_{a \sim \pi'(\cdot|s)} [A^\pi(s, a)] \leq V^{\pi'}(\mu) - V^\pi(\mu) \leq \frac{1}{1-\gamma} \mathbb{E}_{s \sim d_\mu^{\pi',P}} \mathbb{E}_{a \sim \pi'(\cdot|s)} [A^\pi(s, a)]. \quad (11)$$

*Proof.* Equation (11) is given by Lemma 8 from Zhou et al. (2024). $\square$

*Remark* C.6. Moreover, we have

$$V^{\pi'}(\mu) - V^\pi(\mu) \leq \frac{1}{1-\gamma} \mathbb{E}_{s \sim d_\mu^{\pi',P}} \mathbb{E}_{a \sim \pi'(\cdot|s)} [A^\pi(s, a)]$$

$$\leq \frac{1}{1-\gamma} \mathbb{E}_{s \sim d_\mu^{\pi',P}} \mathbb{E}_{a \sim \pi'(\cdot|s)} [A^\pi(s, a)] - \frac{1}{1-\gamma} \mathbb{E}_{s \sim d_\mu^{\pi',P'}} \mathbb{E}_{a \sim \pi'(\cdot|s)} [A^\pi(s, a)]$$

$$+ \frac{1}{1-\gamma}\mathbb{E}_{s \sim d_\mu^{\pi',P'}}\mathbb{E}_{a \sim \pi'(\cdot|s)}[A^\pi(s,a)]$$

$$\leq \frac{1}{1-\gamma}\mathbb{E}_{s \sim d_\mu^{\pi',P'}}\mathbb{E}_{a \sim \pi'(\cdot|s)}[A^\pi(s,a)] + \frac{2}{(1-\gamma)^2}d_{\text{TV}}(d_\mu^{\pi',P'}, d_\mu^{\pi',P})$$

**Lemma C.7.** *Let the NPG update rule be given by*

$$\pi_{t+1}(a|s) = \pi_t(a|s)\frac{\exp\left(\eta\hat{Q}^{\pi_t}(s,a)/(1-\gamma)\right)}{Z_t},$$

*where the normalization factor $Z_t := \sum_{a \in \mathcal{A}}\pi_t(a|s)\exp\left(\eta\hat{Q}_t(s,a)/(1-\gamma)\right)$. Then*

$$\hat{Q}^{\pi_t}(s,a) = \frac{1-\gamma}{\eta}\log Z_t\frac{\pi_{t+1}(a|s)}{\pi_t(a|s)}.$$

*Proof.* See Lemma 4 from Xu et al. (2021). $\qquad\square$

The following lemma tells how much the worst-case value function of a given reward $r_i$ is improved by updating $\pi_t$ to $\pi_{t+1}$ using its corresponding robust policy gradient. The first term $\frac{1}{\eta}\mathbb{E}_{s \sim d_\nu^{\pi_{t+1},P'}}[d_{\text{KL}}\left(\pi_{t+1}(\cdot|s)\|\pi_t(\cdot|s)\right)]$ is always non-negative. The second term $\Delta_t$ will be merged with other errors later.

**Lemma C.8.** *Under the NPG update rule with learning rate $\eta$, the robust value functions with an arbitrary initial distribution $\nu$ satisfy the following inequality:*

$$V_{i_t}^{\pi_{t+1}}(\nu) - V_{i_t}^{\pi_t}(\nu) \geq \frac{1}{\eta}\mathbb{E}_{s \sim d_\nu^{\pi_{t+1},P'}}[d_{KL}\left(\pi_{t+1}(\cdot|s)\|\pi_t(\cdot|s)\right)] + \Delta_t,$$

*where the worst-case transition probability of the policy $\pi_t$ and $\pi_{t+1}$ are $P$ and $P'$, respectively, and the error term $\Delta_t$ is given by*

$$\Delta_t := \frac{1}{1-\gamma}\mathbb{E}_{s \sim d_\nu^{\pi_{t+1},P'}}\sum_{a \in \mathcal{A}}[\pi_t(a \mid s) - \pi_{t+1}(a \mid s)]\left(Q_{i_t}^{\pi_t}(s,a) - \hat{Q}_{i_t}^{\pi_t}(s,a)\right)$$

$$+ \frac{(1-\gamma)}{\eta}\mathbb{E}_{s \sim \nu}[\log Z_t] - \mathbb{E}_{s \sim \nu}[V_{i_t}^{\pi_t}(s)] - \mathbb{E}_{s \sim \nu}\sum_{a \in \mathcal{A}}\pi_t(a \mid s)\left(\hat{Q}_{i_t}^{\pi_t}(s,a) - Q_{i_t}^{\pi_t}(s,a)\right).$$

*Proof.* Let the worst-case transition probability of the policy $\pi_t$ and $\pi_{t+1}$ be $P$ and $P'$, respectively. Their corresponding visitation probabilities are $d_\nu^{\pi_t,P}(s)$ and $d_\nu^{\pi_{t+1},P'}(s)$. Applying Lemma C.5 to the robust value function $V_{i_t}^{\pi_t}(\nu)$ and $V_{i_t}^{\pi_{t+1}}(\nu)$, we obtain

$$V_{i_t}^{\pi_{t+1}}(\nu) - V_{i_t}^{\pi_t}(\nu) \geq \frac{1}{1-\gamma}\mathbb{E}_{s \sim d_\nu^{\pi_{t+1},P'}}\sum_{a \in \mathcal{A}}\pi_{t+1}(a \mid s)A_{i_t}^{\pi_t}(s,a)$$

$$= \frac{1}{1-\gamma}\mathbb{E}_{s \sim d_\nu^{\pi_{t+1},P'}}\sum_{a \in \mathcal{A}}\pi_{t+1}(a \mid s)[Q_{i_t}^{\pi_t}(s,a) - V_{i_t}^{\pi_t}(s)]$$

$$= \frac{1}{1-\gamma}\mathbb{E}_{s \sim d_\nu^{\pi_{t+1},P'}}\sum_{a \in \mathcal{A}}\pi_{t+1}(a \mid s)[\hat{Q}_{i_t}^{\pi_t}(s,a)]$$

$$+ \frac{1}{1-\gamma}\mathbb{E}_{s \sim d_\nu^{\pi_{t+1},P'}}\sum_{a \in \mathcal{A}}\pi_{t+1}(a \mid s)[Q_{i_t}^{\pi_t}(s,a) - \hat{Q}_{i_t}^{\pi_t}(s,a)]$$

$$- \frac{1}{1-\gamma}\mathbb{E}_{s \sim d_\nu^{\pi_{t+1},P'}}[V_{i_t}^{\pi_t}(s)].$$

where the first equality applies the definition of the worst-case advantage function $A^\pi(s,a) := Q^\pi(s,a) - V^\pi(s)$, and the second equality applies the decomposition of the Q-function with its approximation error. By the NPG update rule (Lemma C.7), we have

$$\hat{Q}_{i_t}^{\pi_t}(s,a) = \frac{1-\gamma}{\eta}\log Z_t \frac{\pi_{t+1}(a|s)}{\pi_t(a|s)}.$$

Then we obtain

$$
\begin{aligned}
V_{i_t}^{\pi_{t+1}}(\nu) - V_{i_t}^{\pi_t}(\nu) &\geq \frac{1}{1-\gamma}\mathbb{E}_{s\sim d_\nu^{\pi_{t+1}},P'}\sum_{a\in\mathcal{A}}\pi_{t+1}(a\mid s)\left[\frac{1-\gamma}{\eta}\log Z_t\frac{\pi_{t+1}(a|s)}{\pi_t(a|s)}\right] \\
&\quad + \frac{1}{1-\gamma}\mathbb{E}_{s\sim d_\nu^{\pi_{t+1}},P'}\sum_{a\in\mathcal{A}}\pi_{t+1}(a\mid s)[Q_{i_t}^{\pi_t}(s,a)-\hat{Q}_{i_t}^{\pi_t}(s,a)] \\
&\quad - \frac{1}{1-\gamma}\mathbb{E}_{s\sim d_\nu^{\pi_{t+1}},P'}[V_{i_t}^{\pi_t}(s)] \\
&= \frac{1}{\eta}\mathbb{E}_{s\sim d_\nu^{\pi_{t+1}},P'}\sum_{a\in\mathcal{A}}\pi_{t+1}(a\mid s)\left[\log Z_t + \log\frac{\pi_{t+1}(a|s)}{\pi_t(a|s)}\right] \\
&\quad + \frac{1}{1-\gamma}\mathbb{E}_{s\sim d_\nu^{\pi_{t+1}},P'}\sum_{a\in\mathcal{A}}\pi_{t+1}(a\mid s)[Q_{i_t}^{\pi_t}(s,a)-\hat{Q}_{i_t}^{\pi_t}(s,a)] \\
&\quad - \frac{1}{1-\gamma}\mathbb{E}_{s\sim d_\nu^{\pi_{t+1}},P'}[V_{i_t}^{\pi_t}(s)] \\
&\stackrel{(i)}{=} \frac{1}{\eta}\mathbb{E}_{s\sim d_\nu^{\pi_{t+1}},P'}d_{\mathrm{KL}}(\pi_{t+1}(\cdot|s)\|\pi_t(\cdot|s)) + \frac{1}{\eta}\mathbb{E}_{s\sim d_\nu^{\pi_{t+1}},P'}\log Z_t \\
&\quad + \frac{1}{1-\gamma}\mathbb{E}_{s\sim d_\nu^{\pi_{t+1}},P'}\sum_{a\in\mathcal{A}}\pi_{t+1}(a\mid s)[Q_{i_t}^{\pi_t}(s,a)-\hat{Q}_{i_t}^{\pi_t}(s,a)] \\
&\quad - \frac{1}{1-\gamma}\mathbb{E}_{s\sim d_\nu^{\pi_{t+1}},P'}[V_{i_t}^{\pi_t}(s)].
\end{aligned}
$$

where (i) applies the definition of KL-divergence $d_{\mathrm{KL}}(\pi_{t+1}(\cdot|s)\|\pi_t(\cdot|s)) = \sum_{a\in\mathcal{A}}\pi_{t+1}(a\mid s)\left[\log\frac{\pi_{t+1}(a|s)}{\pi_t(a|s)}\right]$.

By the definition of $Z_t$, we have

$$
\begin{aligned}
\frac{1}{\eta}\mathbb{E}_{s\sim d_\mu^{\pi_{t+1}},P'}\log Z_t(s) &= \frac{1}{\eta}\mathbb{E}_{s\sim d_\mu^{\pi_{t+1}},P'}\log\sum_{a\in\mathcal{A}}\pi_t(a|s)\exp\left(\eta\hat{Q}_{i_t}^{\pi_t}(s,a)/(1-\gamma)\right) \\
&\stackrel{(i)}{\geq} \frac{1}{\eta}\mathbb{E}_{s\sim d_\mu^{\pi_{t+1}},P'}\sum_{a\in\mathcal{A}}\pi_t(a|s)\log\exp\left(\eta\hat{Q}_{i_t}^{\pi_t}(s,a)/(1-\gamma)\right) \\
&= \frac{1}{1-\gamma}\mathbb{E}_{s\sim d_\mu^{\pi_{t+1}},P'}\sum_{a\in\mathcal{A}}\pi_t(a|s)\left(\hat{Q}_{i_t}^{\pi_t}(s,a)\right) \\
&= \frac{1}{1-\gamma}\mathbb{E}_{s\sim d_\mu^{\pi_{t+1}},P'}\sum_{a\in\mathcal{A}}\pi_t(a|s)\left(\hat{Q}_{i_t}^{\pi_t}(s,a)-Q_{i_t}^{\pi_t}(s,a)+Q_{i_t}^{\pi_t}(s,a)\right) \\
&\stackrel{(ii)}{=} \frac{1}{1-\gamma}\mathbb{E}_{s\sim d_\mu^{\pi_{t+1}},P'}\sum_{a\in\mathcal{A}}\pi_t(a|s)\left(\hat{Q}_{i_t}^{\pi_t}(s,a)-Q_{i_t}^{\pi_t}(s,a)\right) + \frac{1}{1-\gamma}\mathbb{E}_{s\sim d_\mu^{\pi_{t+1}},P'}V_{i_t}^{\pi_t}(s,a),
\end{aligned}
$$
(12)

where (i) applies the Jensen's inequality, and (ii) applies the relation between Q-function and value function (Proposition 2.2. from Li et al. (2022)). As the result, we obtain

$$
\begin{aligned}
&V_{i_t}^{\pi_{t+1}}(\nu) - V_{i_t}^{\pi_t}(\nu) \\
&\geq \frac{1}{\eta}\mathbb{E}_{s\sim d_\nu^{\pi_{t+1}},P'}d_{\mathrm{KL}}(\pi_{t+1}(\cdot|s)\|\pi_t(\cdot|s)) + \frac{1}{\eta}\mathbb{E}_{s\sim d_\nu^{\pi_{t+1}},P'}\log Z_t - \frac{1}{1-\gamma}\mathbb{E}_{s\sim d_\nu^{\pi_{t+1}},P'}[V_{i_t}^{\pi_t}(s)]
\end{aligned}
$$

$$- \frac{1}{1-\gamma} \mathbb{E}_{s \sim d_\mu^{\pi_{t+1}, P'}} \sum_{a \in \mathcal{A}} \pi_t(a|s) \left( \hat{Q}_{i_t}^{\pi_t}(s,a) - Q_{i_t}^{\pi_t}(s,a) \right)$$

$$+ \frac{1}{1-\gamma} \mathbb{E}_{s \sim d_\mu^{\pi_{t+1}, P'}} \sum_{a \in \mathcal{A}} \pi_t(a|s) \left( \hat{Q}_{i_t}^{\pi_t}(s,a) - Q_{i_t}^{\pi_t}(s,a) \right)$$

$$+ \frac{1}{1-\gamma} \mathbb{E}_{s \sim d_\nu^{\pi_{t+1}, P'}} \sum_{a \in \mathcal{A}} \pi_{t+1}(a \mid s) [Q_{i_t}^{\pi_t}(s,a) - \hat{Q}_{i_t}^{\pi_t}(s,a)]$$

$$\overset{(i)}{\geq} \frac{1}{\eta} \mathbb{E}_{s \sim d_\nu^{\pi_{t+1}, P'}} d_{\mathrm{KL}}(\pi_{t+1}(\cdot|s) \| \pi_t(\cdot|s)) + \frac{1}{1-\gamma} \mathbb{E}_{s \sim d_\nu^{\pi_{t+1}, P'}} \sum_{a \in \mathcal{A}} \pi_{t+1}(a \mid s) [Q_{i_t}^{\pi_t}(s,a) - \hat{Q}_{i_t}^{\pi_t}(s,a)]$$

$$+ \frac{1}{1-\gamma} \mathbb{E}_{s \sim d_\mu^{\pi_{t+1}, P'}} \sum_{a \in \mathcal{A}} \pi_t(a|s) \left( \hat{Q}_{i_t}^{\pi_t}(s,a) - Q_{i_t}^{\pi_t}(s,a) \right)$$

$$+ \frac{1(1-\gamma)}{\eta} \mathbb{E}_{s \sim \nu} \log Z_t - \mathbb{E}_{s \sim \nu}[V_{i_t}^{\pi_t}(s)] - \mathbb{E}_{s \sim \nu} \sum_{a \in \mathcal{A}} \pi_t(a|s) \left( \hat{Q}_{i_t}^{\pi_t}(s,a) - Q_{i_t}^{\pi_t}(s,a) \right).$$

where (i) we apply the change of measure to replace $d_\nu^{\pi_{t+1}, P'}$ with $\nu$: (1) $\log Z_t(s) + \frac{\eta}{1-\gamma} \sum_{a \in \mathcal{A}} \pi_{t+1}(a \mid s)[Q_{i_t}^{\pi_t}(s,a) - \hat{Q}_{i_t}^{\pi_t}(s,a)] - \frac{\eta}{1-\gamma}[V_{i_t}^{\pi_t}(s)] \geq 0$ for all $s$ by Equation (12), and (2) $\frac{d_\nu^{\pi_{t+1}, P'}}{\nu}(s) \geq 1 - \gamma$. Therefore, we conclude that

$$V_{i_t}^{\pi_{t+1}}(\nu) - V_{i_t}^{\pi_t}(\nu)$$

$$\geq \frac{1}{\eta} \mathbb{E}_{s \sim d_\nu^{\pi_{t+1}, P'}} d_{\mathrm{KL}}(\pi_{t+1}(\cdot|s) \| \pi_t(\cdot|s)) + \frac{1}{1-\gamma} \mathbb{E}_{s \sim d_\nu^{\pi_{t+1}, P'}} \sum_{a \in \mathcal{A}} \pi_{t+1}(a \mid s) [Q_{i_t}^{\pi_t}(s,a) - \hat{Q}_{i_t}^{\pi_t}(s,a)]$$

$$+ \frac{1}{1-\gamma} \mathbb{E}_{s \sim d_\mu^{\pi_{t+1}, P'}} \sum_{a \in \mathcal{A}} \pi_t(a|s) \left( \hat{Q}_{i_t}^{\pi_t}(s,a) - Q_{i_t}^{\pi_t}(s,a) \right)$$

$$+ \frac{(1-\gamma)}{\eta} \mathbb{E}_{s \sim \nu}[V_{i_t}^{\pi_t}(s)] - \mathbb{E}_{s \sim \nu}[V_{i_t}^{\pi_t}(s)] - \mathbb{E}_{s \sim \nu} \sum_{a \in \mathcal{A}} \pi_t(a|s) \left( \hat{Q}_{i_t}^{\pi_t}(s,a) - Q_{i_t}^{\pi_t}(s,a) \right).$$

It completes the proof. $\qquad \square$

The following lemma is the main bound that we will deal with.

**Lemma C.9.** *Under the NPG update rule with learning rate $\eta$, the robust value functions satisfy the following inequality:*

$$V_{i_t}^{\pi^*}(\mu) - V_{i_t}^{\pi_{t+1}}(\mu) \leq \frac{1}{\eta} \left( \mathbb{E}_{s \sim \nu^*} D_{\mathrm{KL}}(\pi^*(\cdot|s) \| \pi_t(\cdot|s)) - \mathbb{E}_{s \sim \nu^*} D_{\mathrm{KL}}(\pi^*(\cdot|s) \| \pi_{t+1}(\cdot|s)) \right)$$

$$+ \frac{1}{1-\gamma} \mathbb{E}_{s \sim \nu^*} \sum_{a \in \mathcal{A}} \pi^*(a \mid s)[Q_{i_t}^{\pi_t}(s,a) - \hat{Q}_{i_t}^{\pi_t}(s,a)]$$

$$+ \frac{1}{(1-\gamma)} \left[ V_{i_t}^{\pi_{t+1}}(\nu^*) - V_{i_t}^{\pi_t}(\nu^*) - \frac{1}{\eta} \mathbb{E}_{s \sim d_{\nu^*}^{\pi_{t+1}, P'}} D_{\mathrm{KL}}(\pi_{t+1}(\cdot|s) \| \pi_t(\cdot|s)) \right.$$

$$- \frac{1}{1-\gamma} \mathbb{E}_{s \sim d_{\nu^*}^{\pi_{t+1}, P'}} \sum_{a \in \mathcal{A}} \pi_{t+1}(a \mid s)[Q_{i_t}^{\pi_t}(s,a) - \hat{Q}_{i_t}^{\pi_t}(s,a)]$$

$$\left. - \frac{1}{1-\gamma} \mathbb{E}_{s \sim d_\mu^{\pi_{t+1}, P'}} \sum_{a \in \mathcal{A}} \pi_t(a|s) \left( \hat{Q}_{i_t}^{\pi_t}(s,a) - Q_{i_t}^{\pi_t}(s,a) \right) \right] + \psi_t,$$

*where the worst-case transition probability of the worst-case optimal policy $\pi^*$ and $\pi_{t+1}$ are $P^*$ and $P'$, respectively, and the visitation probability of $\pi^*$ is $\nu^*(s) := d_\mu^{\pi^*, P^*}(s)$.*

*Proof.* Let the worst-case transition probability of the worst-case optimal policy $\pi^*$ be $P^*$ and the visitation probability of $\pi^*$ be

$$\nu^*(s) := d_\mu^{\pi^*, P^*}(s).$$

Define

$$\psi_t := \frac{1}{1-\gamma}\mathbb{E}_{s\sim d_\mu^{\pi^*},P_t}\mathbb{E}_{a\sim\pi^*(\cdot|s)}[A_{i_t}^{\pi_t}(s,a)] - \frac{1}{1-\gamma}\mathbb{E}_{s\sim d_\mu^{\pi^*},P^*}\mathbb{E}_{a\sim\pi^*(\cdot|s)}[A_{i_t}^{\pi_t}(s,a)], \tag{13}$$

where $P_t$ denotes the worst-case transition probability of $\pi_t$.

Applying Lemma C.5 to the robust value function $V_{i_t}^{\pi^*}$ and $V_{i_t}^{\pi_t}$ $(i = 0, 1, \ldots, I)$, we obtain

$$
\begin{aligned}
V_{i_t}^{\pi^*}(\mu) - V_{i_t}^{\pi_t}(\mu) &\le \frac{1}{1-\gamma}\mathbb{E}_{s\sim\nu^*}\sum_{a\in\mathcal{A}}\pi^*(a\mid s)A_{i_t}^{\pi_t}(s,a)+\psi_t \\
&= \frac{1}{1-\gamma}\mathbb{E}_{s\sim\nu^*}\sum_{a\in\mathcal{A}}\pi^*(a\mid s)[Q_{i_t}^{\pi_t}(s,a) - V_{i_t}^{\pi_t}(s)]+\psi_t \\
&= \frac{1}{1-\gamma}\mathbb{E}_{s\sim\nu^*}\sum_{a\in\mathcal{A}}\pi^*(a\mid s)[\hat{Q}_{i_t}^{\pi_t}(s,a)] + \frac{1}{1-\gamma}\mathbb{E}_{s\sim\nu^*}\sum_{a\in\mathcal{A}}\pi^*(a\mid s)[Q_{i_t}^{\pi_t}(s,a) - \hat{Q}_{i_t}^{\pi_t}(s,a)] \\
&\quad - \frac{1}{1-\gamma}\mathbb{E}_{s\sim\nu^*}[V_{i_t}^{\pi_t}(s)]+\psi_t.
\end{aligned}
$$

where the first equality applies the definition of the worst-case advantage function $A^\pi(s,a) := Q^\pi(s,a) - V^\pi(s)$, and the second equality applies the decomposition of the Q-function with its approximation error. By the NPG update rule (Lemma C.7), we have

$$\hat{Q}^{\pi_t}(s,a) = \frac{1-\gamma}{\eta}\log\left(Z_t\frac{\pi_{t+1}(a|s)}{\pi_t(a|s)}\right).$$

Then we obtain

$$
\begin{aligned}
V_{i_t}^{\pi^*}(\mu) - V_{i_t}^{\pi_t}(\mu) &\le \frac{1}{1-\gamma}\mathbb{E}_{s\sim\nu^*}\sum_{a\in\mathcal{A}}\pi^*(a\mid s)\left[\frac{1-\gamma}{\eta}\log\left(Z_t\frac{\pi_{t+1}(a|s)}{\pi_t(a|s)}\right)\right] \\
&\quad + \frac{1}{1-\gamma}\mathbb{E}_{s\sim\nu^*}\sum_{a\in\mathcal{A}}\pi^*(a\mid s)[Q_{i_t}^{\pi_t}(s,a) - \hat{Q}_{i_t}^{\pi_t}(s,a)] \\
&\quad - \frac{1}{1-\gamma}\mathbb{E}_{s\sim\nu^*}[V_{i_t}^{\pi_t}(s)]+\psi_t \\
&= \frac{1}{\eta}\mathbb{E}_{s\sim\nu^*}\left[\log Z_t + \sum_{a\in\mathcal{A}}\pi^*(a\mid s)\log\frac{\pi_{t+1}(a|s)}{\pi_t(a|s)}\right] \\
&\quad + \frac{1}{1-\gamma}\mathbb{E}_{s\sim\nu^*}\sum_{a\in\mathcal{A}}\pi^*(a\mid s)[Q_{i_t}^{\pi_t}(s,a) - \hat{Q}_{i_t}^{\pi_t}(s,a)] \\
&\quad - \frac{1}{1-\gamma}\mathbb{E}_{s\sim\nu^*}[V_{i_t}^{\pi_t}(s)]+\psi_t \\
&\overset{(i)}{=} \frac{1}{\eta}\mathbb{E}_{s\sim\nu^*}\left[\log Z_t + d_{\mathrm{KL}}\left(\pi^*(\cdot|s)\|\pi_t(\cdot|s)\right) - d_{\mathrm{KL}}\left(\pi^*(\cdot|s)\|\pi_{t+1}(\cdot|s)\right)\right] \\
&\quad + \frac{1}{1-\gamma}\mathbb{E}_{s\sim\nu^*}\sum_{a\in\mathcal{A}}\pi^*(a\mid s)[Q_{i_t}^{\pi_t}(s,a) - \hat{Q}_{i_t}^{\pi_t}(s,a)] - \frac{1}{1-\gamma}\mathbb{E}_{s\sim\nu^*}[V_{i_t}^{\pi_t}(s)]+\psi_t \\
&= \frac{1}{\eta}\mathbb{E}_{s\sim\nu^*}\log Z_t + \frac{1}{\eta}\mathbb{E}_{s\sim\nu^*}\left[d_{\mathrm{KL}}\left(\pi^*(\cdot|s)\|\pi_t(\cdot|s)\right) - d_{\mathrm{KL}}\left(\pi^*(\cdot|s)\|\pi_{t+1}(\cdot|s)\right)\right] \\
&\quad + \frac{1}{1-\gamma}\mathbb{E}_{s\sim\nu^*}\sum_{a\in\mathcal{A}}\pi^*(a\mid s)[Q_{i_t}^{\pi_t}(s,a) - \hat{Q}_{i_t}^{\pi_t}(s,a)] - \frac{1}{1-\gamma}\mathbb{E}_{s\sim\nu^*}[V_{i_t}^{\pi_t}(s)]+\psi_t
\end{aligned}
$$

where (i) applies the definition of KL-divergence. By Lemma C.8, we have

$$\frac{(1-\gamma)}{\eta}\mathbb{E}_{s\sim\nu^*}\log Z_t - \mathbb{E}_{s\sim\nu^*}[V_{i_t}^{\pi_t}(s)] - \mathbb{E}_{s\sim\nu^*}\sum_{a\in\mathcal{A}}\pi_t(a|s)\left(\hat{Q}_{i_t}^{\pi_t}(s,a) - Q_{i_t}^{\pi_t}(s,a)\right)$$

$$\leq V_{i_t}^{\pi_{t+1}}(\nu^*) - V_{i_t}^{\pi_t}(\nu^*) - \frac{1}{\eta}\mathbb{E}_{s\sim d_{\nu^*}^{\pi_{t+1}},P'}d_{\mathrm{KL}}(\pi_{t+1}(\cdot|s)\|\pi_t(\cdot|s))$$

$$- \frac{1}{1-\gamma}\mathbb{E}_{s\sim d_{\nu^*}^{\pi_{t+1}},P'}\sum_{a\in\mathcal{A}}\pi_{t+1}(a\mid s)[Q_{i_t}^{\pi_t}(s,a)-\hat{Q}_{i_t}^{\pi_t}(s,a)]$$

$$- \frac{1}{1-\gamma}\mathbb{E}_{s\sim d_{\mu}^{\pi_{t+1}},P'}\sum_{a\in\mathcal{A}}\pi_t(a|s)\left(\hat{Q}_{i_t}^{\pi_t}(s,a)-Q_{i_t}^{\pi_t}(s,a)\right)$$

Then we obtain

$$V_{i_t}^{\pi^*}(\mu) - V_{i_t}^{\pi_{t+1}}(\mu) \leq \frac{1}{\eta}\left(\mathbb{E}_{s\sim\nu^*}D_{\mathrm{KL}}(\pi^*(\cdot|s)\|\pi_t(\cdot|s)) - \mathbb{E}_{s\sim\nu^*}D_{\mathrm{KL}}(\pi^*(\cdot|s)\|\pi_{t+1}(\cdot|s))\right)$$

$$+ \frac{1}{1-\gamma}\mathbb{E}_{s\sim\nu^*}\sum_{a\in\mathcal{A}}\pi^*(a\mid s)[Q_{i_t}^{\pi_t}(s,a)-\hat{Q}_{i_t}^{\pi_t}(s,a)]$$

$$+ \frac{1}{(1-\gamma)}\left[V_{i_t}^{\pi_{t+1}}(\nu^*) - V_{i_t}^{\pi_t}(\nu^*) - \frac{1}{\eta}\mathbb{E}_{s\sim d_{\nu^*}^{\pi_{t+1}},P'}D_{\mathrm{KL}}(\pi_{t+1}(\cdot|s)\|\pi_t(\cdot|s))\right.$$

$$- \frac{1}{1-\gamma}\mathbb{E}_{s\sim d_{\nu^*}^{\pi_{t+1}},P'}\sum_{a\in\mathcal{A}}\pi_{t+1}(a\mid s)[Q_{i_t}^{\pi_t}(s,a)-\hat{Q}_{i_t}^{\pi_t}(s,a)]$$

$$\left.- \frac{1}{1-\gamma}\mathbb{E}_{s\sim d_{\mu}^{\pi_{t+1}},P'}\sum_{a\in\mathcal{A}}\pi_t(a|s)\left(\hat{Q}_{i_t}^{\pi_t}(s,a)-Q_{i_t}^{\pi_t}(s,a)\right)\right]+\psi_t$$

which is the final upper bound after applying the last inequality. Then we omit the term containing $-D_{\mathrm{KL}}(\pi_{t+1}(\cdot|s)\|\pi_t(\cdot|s))$ since it is always non-positive. □

The following lemma gives the condition such that $\mathcal{N}_0$ is non-empty.

**Lemma C.10.** *Consider the NPG update rule with learning rate $\eta$ and let $\delta > 0$ be chosen such that*

$$\delta > \frac{1}{T}\mathbb{E}_{s\sim\nu^*}D_{\mathrm{KL}}\left(\pi^*(\cdot|s)\|\pi_1(\cdot|s)\right) + \frac{\eta L}{(1-\gamma)^2} + \bar{\epsilon}_{approx} + \frac{2\ell}{(1-\gamma)^2}c,$$

*where $\bar{\epsilon}_{approx}$ is an error term depending on the approximation error terms, $\ell$ is the Lipschitz constant of the state visitation distribution $d_{\nu}^{\pi,P}$, and $L$ is the Lipschitz constant of the robust value function $V_i^\pi(\nu^*)$, $c$ is the diameter of the uncertainty set given by Assumption C.4. Under these conditions,*

$$\mathcal{N}_0 := \{t : V_i^{\pi^*}(\mu) - V_i^{\pi_t}(\mu) \geq d_i - \delta \text{ for all } i\}$$

*is always non-empty.*

*Proof.* When $t \in \mathcal{N}_i := \{t : V_i^{\pi_t} \text{ is sampled to update}\}$, we have $i_t = i$ and by the algorithm design,

$$V_i^{\pi^*}(\mu) - V_i^{\pi_{t+1}}(\mu) \geq V_i^{\pi^*}(\mu) - d_i + \delta + \hat{V}_i^{\pi_{t+1}}(\mu) - V_i^{\pi_{t+1}}(\mu)$$

$$\geq \delta - \left[\hat{V}_i^{\pi_{t+1}}(\mu) - V_i^{\pi_{t+1}}(\mu)\right].$$

We sum the inequality obtained from Lemma C.9 over $t = 1, 2, \ldots, T$. Since the robust value function $V^\pi(\mu)$ is Lipschitz in $\pi$ (Wang & Zou, 2021; Zhou et al., 2024), we have

$$|V^{\pi_{t+1}}(\nu^*) - V^{\pi_t}(\nu^*)| \leq L\|\pi_{t+1} - \pi_t\| \leq \frac{L\eta}{1-\gamma}.$$

Then we obtain

$$\eta\sum_{i\in\mathcal{N}_0}\left(V_i^{\pi^*}(\mu) - V_i^{\pi_{t+1}}(\mu)\right) + \eta\delta T \leq \eta\mathbb{E}_{s\sim\nu^*}D_{\mathrm{KL}}(\pi^*(\cdot|s)\|\pi_1(\cdot|s)) + \frac{\eta^2 LT}{(1-\gamma)^2} + \eta T\bar{\epsilon}_{approx} + \eta\sum_{t=1}^{T}\psi_t,$$

where $\psi_t$ is given by Equation (13) and $\bar{\epsilon}_{\text{approx}}$ is a constant upper bound (depending on Assumption 4.1) of $C_{\text{approx}}$ which is defined as

$$
\begin{aligned}
C_{\text{approx}} := &\frac{1}{1-\gamma} \mathbb{E}_{s \sim \nu^*} \sum_{a \in \mathcal{A}} \pi^*(a \mid s)[Q_i^{\pi_t}(s,a) - \hat{Q}_i^{\pi_t}(s,a)] \\
&+ \frac{1}{(1-\gamma)} \left[ -\left[\hat{V}_i^{\pi_{t+1}}(\mu) - V_i^{\pi_{t+1}}(\mu)\right] - \frac{1}{1-\gamma} \mathbb{E}_{s \sim d_{\nu^*}^{\pi_{t+1},P'}} \sum_{a \in \mathcal{A}} \pi_{t+1}(a \mid s)[Q_i^{\pi_t}(s,a) - \hat{Q}_i^{\pi_t}(s,a)] \right. \\
&\left. - \frac{1}{1-\gamma} \mathbb{E}_{s \sim d_{\mu}^{\pi_{t+1},P'}} \sum_{a \in \mathcal{A}} \pi_t(a|s) \left( \hat{Q}_i^{\pi_t}(s,a) - Q_i^{\pi_t}(s,a) \right) \right].
\end{aligned}
$$

By appropriately choosing the policy evaluation algorithm (discussed in Appendix C.3), $\bar{\epsilon}_{\text{approx}}$ can be arbitrarily small. If $\mathcal{N}_0 = \emptyset$, then

$$
\eta \delta T \leq \eta \mathbb{E}_{s \sim \nu^*} D_{\text{KL}}(\pi^*(\cdot|s)\|\pi_1(\cdot|s)) + \frac{\eta^2 LT}{(1-\gamma)^2} + \eta T \bar{\epsilon}_{\text{approx}} + \eta \sum_{t=1}^{T} \psi_t.
$$

We further take

$$
\psi_t \leq \frac{2}{(1-\gamma)^2} d_{\text{TV}}(d_{\mu}^{\pi',P'}, d_{\mu}^{\pi',P}) \leq \frac{2\ell}{(1-\gamma)^2} \sup_{s,a} d_{\text{TV}}(P'(\cdot|s,a), P(\cdot|s,a)) \leq \frac{2\ell}{(1-\gamma)^2} c,
$$

where we apply the $\ell$-Lipschitzness of the state visitation distribution (Lemma 5, Eq. (37), Chen & Huang (2024)). Then, we let

$$
\delta > \frac{1}{T} \mathbb{E}_{s \sim \nu^*} D_{\text{KL}}(\pi^*(\cdot|s)\|\pi_1(\cdot|s)) + \frac{\eta L}{(1-\gamma)^2} + \bar{\epsilon}_{\text{approx}} + \frac{2\ell}{(1-\gamma)^2} c.
$$

This hyper-parameter setting ensures that $\mathcal{N}_0$ is non-empty. $\qquad\square$

## C.3 Robust Policy Evaluation

In this section, we collect two important robust policy evaluation techniques to discuss how to use these methods to obtain the robust value function with sufficient accuracy. Though we use a simplified result in this subsection, these results have been extended to more general setting in original sources.

### C.3.1 Option 1: The IPM Uncertainty Set

**Lemma C.11** (Theorem 3, Zhou et al. (2024)). *Let the value function $V^\pi$ is parameterized by $w \in \mathbb{R}^{|\mathcal{S}|}$ with the linear feature $\phi \in \mathbb{R}^{|\mathcal{S}|}$. Then using the Robust Linear TD-Learning proposed by Zhou et al. (2024) with step sizes $\alpha_k = \Theta(1/k)$, the output satisfies $\mathbb{E}\|w_K - w^*\|^2 = \widetilde{O}(\frac{1}{K})$.*

As shown by Li et al. (2022), the robust Q-function can be calculated using the robust value function learned by the robust TD-learning algorithm described above. That is,

$$
Q^\pi(s,a) = r(s,a) + \gamma \inf_{P \in \mathcal{P}} V^\pi(s').
$$

The second term $\inf_{P \in \mathcal{P}} V^\pi(s')$ is given by Proposition 1 from Zhou et al. (2024). This result indicates that we can obtain the robust Q-function with the convergence rate $\frac{1}{\sqrt{K}}$ (for the $L_\infty$-norm).

### C.3.2 Option 2: The p-Norm Uncertainty Set

We consider the following uncertainty set:

$$
\mathcal{V} := \{v \in R^{|S|} \mid \langle v, 1_{|S|} \rangle = 0, \|v\|_p \leq \beta\},
$$

$$\mathcal{U} := \mathcal{V} + P_0.$$

Let $q$ satisfy $\frac{1}{q} + \frac{1}{p} = 1$. Then $\mathcal{O}_{\beta,p}(\cdot) : R^{|S|} \to R^{|S|}$ is defined as:

$$\mathcal{O}_{\beta,p}(V)(s') := \beta \frac{\text{sign}(V(s') - \omega_q(V))|V(s') - \omega_q(V)|^{q-1}}{\kappa_q(V)^{q-1}},$$

where $\omega_q(V) := \arg\min_\omega \|V - \omega 1_{|S|}\|_q$ and $\kappa_q(V) := \min_\omega \|V - \omega 1_{|S|}\|_q$.

**Lemma C.12** (Theorem 4.2, Kumar et al. (2023)). *If the uncertainty set is defined as the $p$-norm $(s,a)$-rectangular set, then the worst-case transition probability $P_+(\cdot|s,a)$ can be represented as*

$$P_+(\cdot|s,a) = P_0(\cdot|s,a) - \beta \mathcal{O}_{\beta,p}(V)$$

*where $\beta$ is the radius of the uncertainty set and $\mathcal{O}_{\beta,p}(V)$ is the balanced robust value function (Kumar et al., 2023).*

Based on this result, we apply the following TD-learning update rule:

$$V(s) \leftarrow V(s) + \alpha \left( \underbrace{r(s,a) + \gamma V(s') - V(s)}_{\text{stand. TD err. under } P_0} - \gamma \mathcal{O}_{\beta,p}(V)(s')V(s') \right). \tag{14}$$

Here $\mathcal{O}_{\beta,p}(\cdot) : R^{|S|} \to R^{|S|}$ is an operator determined by the uncertainty set. It is easy to observe that this update rule is equivalent to the TD-learning over the worst-case transition probability:

$$
\begin{aligned}
&E_{s' \sim P_0(s'|s,a)}[r(s,a) + \gamma V(s')] - \gamma \langle \mathcal{O}_{\beta,p}(V^\pi), V \rangle \\
=& r(s,a) + \gamma \sum_{s',a} P_0(s'|s,a)\pi(a|s)V(s') - \gamma \sum_{s'} \mathcal{O}_{\beta,p}(V^\pi)(s')V(s') \\
=& r(s,a) + \gamma \sum_{s',a} P_0(s'|s,a)\pi(a|s)V(s') - \gamma \sum_{s',a} \pi(a|s)\mathcal{O}_{\beta,p}(V^\pi)(s')V(s') \\
=& r(s,a) + \gamma \sum_{s',a} \pi(a|s)[P_0(s'|s,a) - \mathcal{O}_{\beta,p}(V_0^\pi)(s')]V(s') \\
\overset{(i)}{=}& r(s,a) + \gamma \sum_{s',a} \pi(a|s)P_+(s'|s,a)V(s') \\
=& r(s,a) + \gamma E_{s' \sim P_+(s'|s,a)}V(s'),
\end{aligned}
$$

where (i) applies the remarkable result from Theorem 4.2, Kumar et al. (2023): the worst-case transition $P_+$ is the rank-one perturbation of the nominal transition $P_0$. Therefore, by applying existing TD-learning convergence analysis (Brandfonbrener & Bruna, 2019; Asadi et al., 2024; Li et al., 2024), we obtain that the convergence rate is also $\frac{1}{\sqrt{K}}$.

## C.4 A Compact Algorithm

We also include a compact version of Algorithm 1 in Algorithm 3 which merges the constraint rectification and objective rectification together.

## C.5 The Proof of Main Theorem

Here, we state the full version of Theorem 4.5.

**Theorem C.13.** *Consider the NPG update rule with learning rate $\eta$. Let the constraint violation tolerance $\delta > 0$ be chosen to satisfy*

$$\delta > \frac{2}{T}\mathbb{E}_{s \sim \nu^*} D_{\text{KL}}\big(\pi^*(\cdot|s)\|\pi_1(\cdot|s)\big) \;+\; \frac{2\eta L}{(1-\gamma)^2} \;+\; 2\bar{\epsilon}_{approx} + \frac{4\ell}{(1-\gamma)^2}c,$$

---

**Algorithm 3:** Rectified Robust Policy Optimization

---

**input** : initial policy parameters $\theta_0$, empty set $\mathcal{N}_0$

**for** $t = 0, \cdots, T-1$ **do**

    Evaluate value functions under $\pi_t := \pi_{\theta_t}$: $\hat{Q}_i^{\pi_t}(s,a) \approx Q_i^{\pi_t}(s,a)$ for $i = 0, 1, \ldots, I$ ;

    Sample state-action pairs $(s_j, a_j)$ from the nominal distribution ;

    Compute value estimates $V_i^{\pi_t}$ for $i = 0, \ldots, I$ ;

    **if** $V_i^{\pi_t} \geq d_i - \delta$ *for all* $i = 0, 1, \ldots, I$ **then**

        // Threshold Updates

        Add $\theta_t$ to set $\mathcal{N}_0$ and track the feasible policy achieving the largest value $\pi_{\text{out}} = \pi_t$;

        Update $d_0$: $d_0^{t+1} \leftarrow V_0^{\pi_t}$;

    **else if** $V_i^{\pi_t} < d_i - \delta$ *for some* $i = 1, \ldots, I, 0$ **then**

        // Constraint Rectification & Objective Rectification

        Maximize $V_i^{\pi_t}$ using Equation (8);

**output:** $\pi_{\text{out}}$

---

*where $\ell$ is the Lipschitz constant of the state visitation distribution $d_\nu^{\pi,P}$, and $L$ is the Lipschitz constant of the robust value function $V_i^\pi(\nu^*)$, $\bar{\epsilon}_{approx}$ is the error caused by the robust policy evaluation step. Under these conditions, the output policy $\pi_{out}$ satisfies:*

$$\mathbb{E}\left[V^*(\mu) - V^{\pi_{out}}(\mu)\right] \leq \frac{2}{T}\mathbb{E}_{s \sim \nu^*}D_{\mathrm{KL}}\left(\pi^*(\cdot|s) \| \pi_1(\cdot|s)\right) + \frac{2\eta L}{(1-\gamma)^2} + 2\bar{\epsilon}_{approx} + \frac{4\ell}{(1-\gamma)^2}c,$$

*where the constraint violation of $\pi_{out}$ is guaranteed to be at most $\delta$. Moreover, if setting*

$$\frac{(1-\gamma)^2}{2L}\frac{\epsilon}{4} \leq \eta \leq \frac{(1-\gamma)^2}{2L}\frac{\epsilon}{3},$$

*the robust policy evaluation error $\epsilon_{approx} \leq \frac{(1-\gamma)^2}{12}\epsilon$, and the number of iteration step*

$$T \geq \frac{2L}{(1-\gamma)^2}\frac{12}{\epsilon^2}\mathbb{E}_{s \sim \nu^*}D_{\mathrm{KL}}(\pi^*(\cdot|s) \| \pi_1(\cdot|s)),$$

*then the output policy satisfies the $\epsilon$-accuracy with a constant error; that is*

$$\mathbb{E}\left[V^*(\mu) - V^{\pi_{out}}(\mu)\right] \leq \epsilon + \frac{4\ell}{(1-\gamma)^2}c.$$

*Proof.* By the update rule, the boundary value $d_0$ is non-decreasing. Since it is upper bounded, we conclude that $\{d_0^t\}$ converges and we denote

$$d_0^t \to \bar{d}_0$$

as $t \to \infty$. More explicitly, we have

$$\bar{d}_0 = \sup\{V_0^{\pi_t} : V_i^{\pi_t} < d_i + \delta\}.$$

There are only two cases for the output policy $\pi_{\text{out}}$:

(1) The policy is better than the optimal policy while the relaxed constraint is violated; i.e.

$$V_i^{\pi^*}(\mu) - V_i^{\pi_{t+1}}(\mu) \leq 0.$$

(2) The output policy is worse than the optimal policy but upper bounded by $\mathcal{O}(\frac{1}{\sqrt{T}})$. When (1) holds, then it is desired. When (1) doesn't hold (i.e. $V_i^{\pi^*}(\mu) - V_i^{\pi_{t+1}}(\mu) > 0$.), we assume $|\mathcal{N}_0| < \frac{T}{2}$. It implies $\sum_{i=1}^I |\mathcal{N}_i| \geq \frac{T}{2}$. Then we have

$$\frac{1}{2}\eta\delta T \leq \eta\mathbb{E}_{s \sim \nu^*}D_{\mathrm{KL}}(\pi^*(\cdot|s) \| \pi_1(\cdot|s)) + \frac{\eta^2 LT}{(1-\gamma)^2} + \eta T\bar{\epsilon}_{\text{approx}} + \eta\sum_{t=1}^T \psi_t$$

$$\leq \eta \mathbb{E}_{s\sim\nu^*} D_{\mathrm{KL}}(\pi^*(\cdot|s)\|\pi_1(\cdot|s)) + \frac{\eta^2 LT}{(1-\gamma)^2} + \eta T \bar{\epsilon}_{\mathrm{approx}} + \eta T \frac{2\ell}{(1-\gamma)^2} c. \tag{15}$$

Here the last inequality applies the upper bound of $\psi_t$ (see Lemma C.10 for the full derivation and the definition of $\ell$). Then we let

$$\delta > \frac{2}{\eta T}\mathbb{E}_{s\sim\nu^*} D_{\mathrm{KL}}(\pi^*(\cdot|s)\|\pi_1(\cdot|s)) + \frac{2\eta L}{(1-\gamma)^2} + 2\bar{\epsilon}_{\mathrm{approx}} + \frac{4\ell}{(1-\gamma)^2} c.$$

This hyper-parameter setting ensures that Equation (15) does not hold. Therefore, it leads to a contradiction. We obtain $|\mathcal{N}_0| \geq \frac{T}{2}$. In this case, we have

$$0 \leq \mathbb{E}\left[V^*(\mu) - V^{\pi_{\mathrm{out}}}(\mu)\right] \leq \mathbb{E}_{\pi\sim\mathcal{N}_0}\left[V^*(\mu) - V^\pi(\mu)\right]$$
$$\leq \frac{2}{\eta T}\mathbb{E}_{s\sim\nu^*} D_{\mathrm{KL}}(\pi^*(\cdot|s)\|\pi_1(\cdot|s)) + \frac{2\eta L}{(1-\gamma)^2} + 2\bar{\epsilon}_{\mathrm{approx}} + \frac{4\ell}{(1-\gamma)^2} c.$$

Here the non-negativity is because $V^*(\mu)$ is the largest-possible value function over the feasible policy. From the construction of $\mathcal{N}_0$ and the output policy $\pi_{\mathrm{out}}$, they are all feasible policies.

We are interested in obtaining the complexity of the convergence up to a constant value function error $\frac{4\ell}{(1-\gamma)^2} c$ (typically, this term can be controlled by using a sufficiently small uncertainty set). To obtain the sample complexity, we set all three terms to be $\mathcal{O}(\epsilon)$:

- Let $\frac{2\eta L}{(1-\gamma)^2} \leq \frac{\epsilon}{3}$. Then we obtain

$$\eta \leq \frac{(1-\gamma)^2}{2L}\frac{\epsilon}{3}.$$

- To make the last term $2\bar{\epsilon}_{\mathrm{approx}} \leq \frac{\epsilon}{3}$, we set the robust policy evaluation error (Assumption 4.1) to be

$$\|\hat{Q} - Q\|_\infty \leq \epsilon_{\mathrm{approx}}.$$

It leads to

$$\frac{1}{1-\gamma}\epsilon_{\mathrm{approx}} + \frac{1}{(1-\gamma)}\left[\epsilon_{\mathrm{approx}} + \frac{1}{1-\gamma}\epsilon_{\mathrm{approx}} + \frac{1}{1-\gamma}\epsilon_{\mathrm{approx}}\right] \leq \frac{\epsilon}{3}.$$

It solves $\epsilon_{\mathrm{approx}} \leq \frac{(1-\gamma)^2}{12}\epsilon$.

- Let $\frac{2}{\eta T}\mathbb{E}_{s\sim\nu^*} D_{\mathrm{KL}}(\pi^*(\cdot|s)\|\pi_1(\cdot|s)) \leq \frac{\epsilon}{3}$. We obtain

$$T \geq \frac{2}{\eta T}\mathbb{E}_{s\sim\nu^*} D_{\mathrm{KL}}(\pi^*(\cdot|s)\|\pi_1(\cdot|s))\frac{3}{\epsilon}$$
$$\geq \frac{2L}{(1-\gamma)^2}\frac{12}{\epsilon^2}\mathbb{E}_{s\sim\nu^*} D_{\mathrm{KL}}(\pi^*(\cdot|s)\|\pi_1(\cdot|s)).$$

In the second step, we require the learning rate $\eta$ is not too small; that is, we let it larger than $\frac{(1-\gamma)^2}{2L}\frac{\epsilon}{3}$. This result indicate that the iteration complexity is $T = \mathcal{O}(\epsilon^{-2})$, with choosing an appropriate learning rate $\eta = \Theta(\epsilon)$ and the approximation error $\epsilon_{\mathrm{approx}} = \mathcal{O}(\epsilon)$.

$\square$

# D    Experiment Setting

This section outlines the information for replicating our experiments. Throughout our experiments, we include multiple independent runs and report the averaged metrics in our results. Specifically, we used three different random seeds.

### D.1 Hardware Specification and System Environment

We conducted our experiments on a computing desktop running Windows 10 Education, equipped with 3200MHz DDR4 DRAM memory, AMD Ryzen 7 3800X 8-Core, 16-Thread processor, and one NVIDIA GeForce RTX 2070 Super graphics cards. All experiments are executed using Python version 3.10.14.

### D.2 FrozenLake-Like Gridworld Experiment

The reward function is defined as follows:

$$
r_0(s, a, s') = \begin{cases} +1 & \text{if } s' \text{ is the target} \\ -1 & \text{if } s' \text{ is a brown block} \\ -0.1 & \text{otherwise} \end{cases}
$$

and define $r(s, a) := \mathbb{E}_{s'}[r_0(s, a, s')]$. The constraint reward function is defined as:

$$
r_1(s, a, s') = \begin{cases} -1 & \text{if } s' \text{ is out of the boundary} \\ -1 & \text{if } s' \text{ is a brown block} \\ 0 & \text{otherwise} \end{cases}.
$$

Here, we further define the cost function $c(s, a) := -\mathbb{E}_{s'}[r_1(s, a, s')]$ to better distinguish it with the rewards. In this experiment, we require the cost value function $-V_1^\pi(\mu)$ less than 0.2, which means that the agent should avoid hitting the brown block or move out of the box.

We used a discount factor $\gamma = 0.99$. The learning rates for both algorithms are set to 0.0001 and the tolerance for constraint violations is $\delta = 0.01$. The robustness of the environment was simulated by introducing a slipping probability $p = 0.2$ in the test environment, which differs from the deterministic dynamics used during training. For both methods, we run 1M steps.

During the training, we use the neural network taking a 2-dimensional input (the position of the agent) and processes it through a single fully connected layers of size 64 followed by a ReLU activation then fed into a final linear layer that produces 4 logits (four actions: Up, Down, Left, and Right).

### D.3 Mountain Car Experiment

We use the standard Mountain Car environment provided by Towers et al. (2024). Once the car reaches the goal, it is reset to the original starting point. The reward function is defined using the environment's default setting:

$$
r_0(s, a, s') = \begin{cases} 0 & \text{if the agent reaches the goal,} \\ -1 & \text{otherwise,} \end{cases}
$$

and we set $r(s, a) := \mathbb{E}_{s'}[r_0(s, a, s')]$. To emphasize safety, we introduce the constraint reward function:

$$
r_1(s, a, s') = \begin{cases} -1 & \text{if the car's speed exceeds 0.06,} \\ 0 & \text{otherwise,} \end{cases}
$$

and define the cost function $c(s, a) := -\mathbb{E}_{s'}[r_1(s, a, s')]$. In this experiment, we account for environment uncertainty by perturbing the "gravity" parameter from its nominal value 0.0025 to 0.003 in the worst-case scenario. In this experiment, we set the constraint to be $-4$ (i.e. we require $-V_1^\pi(\mu) < 4$). As shown in Figure 3b, both CRPO and RRPO learn a feasible solution.

Given that the MountainCar environment has a continuous state space (i.e., the car's position and velocity), we employ radial basis function (RBF) features to achieve a linear approximation of the policy. Specifically, each state $s$ is first transformed into an RBF feature vector $\phi(s)$, which is then multiplied by the policy parameters $\theta$ (one column per action) to generate logits; these logits are passed through a softmax function

to produce the policy distribution over actions. Additionally, we incorporate an $\epsilon$-greedy strategy with an initial $\epsilon = 0.1$, decaying at a rate of 0.9999, to encourage exploration in the early stages of training.

We set the 2-norm $(s, a)$-rectangular uncertainty set defined by Equation (10). Since the state space is continuous, when evaluating the centered value function, we uniformly sample 100 states from the state space to estimate the mean and the variance value of $V^\pi(s)$. The radius of the $p$-norm uncertainty set is set to be 0.0002. This value is manually tuned from a preset hyper-parameter set $\{0.00001, 0.0001, 0.0002, 0.0003, 0.001\}$. When the radius value is too high, the policy tends to be too conservative; when the radius value is too small, the policy performs similar as the non-robust case.

### D.4 Robust Control Experiment

In this experiment, we adopt the common PPO tricks to stablize the training. Each algorithm was trained for 2 million timesteps using identical hyperparameters to ensure fair comparison: learning rate of $3 \times 10^{-4}$, batch size of 256, discount factor $\gamma = 0.99$, and PPO clip range of 0.2. We incorporated energy-based constraints with a threshold of 0.25, representing a moderate constraint that provides meaningful limitations without being overly restrictive. For the constrained algorithms, Primal-Dual Policy Gradient (Wang et al., 2022) employed a constraint learning rate of $3 \times 10^{-4}$ with Lagrange multiplier initialization of 0.1 and maximum value of 100.0. On the construction of the uncertainty set, we follow Hou et al. (2020)'s robust Actor-Critic algorithm to obtain the desired robust value function approximation. The radius of the uncertainty set is set as 0.001. In our experiment, we ran three independent runs and reported the averaged results in the experimental section.

