# OpenReview forum: "Rectified Robust Policy Optimization for  Model-Uncertain Constrained Reinforcement Learning without Strong Duality"
_TMLR — Accepted by TMLR_

### Review · Reviewer_WgLD · 2025-05-26

**Summary Of Contributions:**

This paper studies the robust constrained MDP. The authors first show strong duality does not hold in general under this setting, and thus conventional primal-dual methods for standard constrained MDP and robust MDP do not apply. A primal-only algorithm is proposed to solve the robust constrained MDP, theoretical analysis and experiments results are provided to show the effectiveness of the proposed algorithm.

**Audience:**

Yes

**Claims And Evidence:**

Yes

**Requested Changes:**

I have the following questions:

1. Until 4.1, the current writing gives me an impression that this work is solving a planning problem where the transition and rewards are known, because the algorithm does not involve data collection. However, the statement in section 4.2 and Appendix C.3 implies that it actually involves an online learning process for robust policy evaluation. In discussions, some results from planning papers are cited, which are used in online learning algorithms developed by existing literature. I am not saying this is wrong, but a clearer version of algorithm design would improve the current manuscript, especially the authors highlight accurate evaluation of Q-function is critical. For example, the author might provide detailed subroutine for policy evaluation, showing how the data is collected, how to estimate the Q-function, etc.

2. About Alg 1, just curious if we can merge the 'Constraint Rectification' and 'Objective Rectification'.

3. The authors claim that the Assumption 4.1 and 4.2 used to develop theoretical guarantee are mild. However, they somehow bypass the most difficult challenges in online DRMDPs - policy evaluation and exploration. For example, under what additional conditions/assumptions and specific estimation procedure can we expect Assumption 4.1 to hold? Assumption 4.2 assumes the visitation probability is always lower bounded under any policy $\pi$: on the one hand this assumption is made on the visitation measure induced under the worst case env, while the exploration happens under the nominal env, right?  How does this assumption relate to exploration? How the algorithm deal with exploration-exploitation trade-off? on the other hand, when the state space is large, the lower bound $p_{\min}$ can be extremely small. In this case, will the assumption, as well as corresponding analysis being invalid? How $p_{\min}$ affect your bound and analysis?

4. There are other works on online distributionally robust RL, for example RMDP with $(s,a)$-rectangularity: 'Distributionally robust reinforcement learning with interactive data collection: Fundamental hardness and near-optimal algorithm', and RMDP with $d$-rectangularity: 'Distributionally robust off-dynamics reinforcement learning: Provable efficiency with linear function approximation' and
'Upper and lower bounds for distributionally robust off-dynamics reinforcement learning'. Discussion on whether the framework studied in the manuscript can incorporate these methodology would strengthen this work. Besides, the current discussion on Robust RL is limited. A relatively comprehensive summary of most recent works in this field is encouraged.

-------
Other minor issues:

Page 16, section A.2 $(1−\alpha)d^{\pi, P}+\alpha d^{\pi, P}=d^{\pi′,P}$

Page 6, $\pi_1$  used without definition.

**Strengths And Weaknesses:**

The hard instance and corresponding analysis showing positive duality gap is interesting. Before section 4.2 the paper is in general clearly written. The experiments, though restricted to simple environments, show the worst case robustness and constraints compliances. However, the description of algorithm and its discussion can be more clear and rigorous. Please see questions below.

---

> ### Author Response · Authors · 2025-06-23
>
> We thank the reviewer for the constructive and thoughtful feedback. We have uploaded the revision. Below we provide point-by-point responses.
>
> * ***Q1:*** Clarify the problem setting, provide a more detailed description of the algorithm design, particularly the subroutine for policy evaluation, and include how data is collected and how the Q-function is estimated?
>
>   ***A1:*** Thank you for pointing it out. We agree that the problem setting and algorithm design should be made clearer. We have revised the manuscript to explicitly restate the problem setting and to clarify the robust policy evaluation subroutine in Section 4.1.
>
> * ***Q2:*** Can merge the 'Constraint Rectification' and 'Objective Rectification' together?
>
>   ***A2:*** Yes, these two steps can be merged together, and we provide a merged version in the appendix of the revised manuscript. We chose to present them separately in the main text to better emphasize the logical sequence of updates: we always rectify constraint violations before adjusting the objective. This separation improves readability and helps ensure that infeasible updates are avoided during policy optimization.
>
> * ***Q3:*** (1) Under what conditions can we expect Assumption 4.1 (Policy Evaluation Accuracy) to hold? (2) Assumption 4.2 is made under the worst-case environment, but exploration occurs under the nominal one, how are they related? (3) When the state space is large and $p_{\min}$ is small, does this affect the validity of the assumption and the analysis?
>
>     ***A1:*** We appreciate the reviewer's thoughtful and detailed questions. We address each point below:
>
>     *(1)* Assumption 4.1 is expected to hold under standard conditions studied in the robust policy evaluation literature. In particular, it holds when the function class used to approximate the Q-function is sufficiently expressive and the environment is sufficiently explored. The specific conditions depend on the choice of robust policy evaluation algorithms and different methods  may impose different technical requirements to ensure accuracy. For example, in [Wang2021], it relies on the R-contamination structure of the uncertainty set; learning the robust value function requires to visits all possible state-action pairs to obtain the desired accuracy. In [Zhou2023], it allows the value function is linearly approximated by a specific basis under the IPM uncertainty set; the visitation assumption could be slightly weaken to the supporting assumption (see Assumption 2 in their paper).
>
>     *(2)\&(3)* You are correct in observing this distinction. Assumption 4.2 is indeed imposed on the visitation distribution induced by the worst-case environment, while exploration takes place under the nominal environment. This assumption de facto weakens some existing assumptions; for example, [Chen2024] (Theorem 2) assumes there exists $p_{\min}>0$ such that $\inf_{s,a,s'} p(s'|s,a)\geq p_{\min}$ for all uncertainty transitions $p$. [Zhou2023] (Assumption 2) assumes that the nominal transition is supported by all transitions in the given uncertainty set. We realize that if we are given the access to the robust policy evaluation oracle (Assumption 1 in our paper), these widely used assumptions could be replaced with requiring the exploration on the worst-case transition instead of each transition in the uncertainty set.
>
>     In addition, we also introduce this assumption for some technical reasons: we rely on a change-of-measure trick to convert expectations over the nominal environment into expectations under the worst-case one. As a result, this mismatch is reflected in our final bounds through the value $C_u$, which scales linearly with $1/p_{\min}$. When the state space is large and $p_{\min}$ is small, $C_u$ can become large, which in turn slows down convergence and increases the number of samples required to achieve a given level of accuracy.
>
>     * [Wang2021] Wang, Yue, and Shaofeng Zou. "Online robust reinforcement learning with model uncertainty." Advances in Neural Information Processing Systems 34 (2021): 7193-7206.
>     * [Zhou2023] Zhou, Ruida, et al. "Natural actor-critic for robust reinforcement learning with function approximation." Advances in neural information processing systems 36 (2023): 97-133.
>     * [Chen2024] Chen, Ziyi, and Heng Huang. "Accelerated policy gradient for s-rectangular robust MDPs with large state spaces." Forty-first International Conference on Machine Learning. 2024.

---

> > ### Author Response · Authors · 2025-06-23
> >
> > * ***Q4:*** Discussions on other related RMDP references.
> >
> >   ***A4:*** We agree that our current version lacks sufficient discussions on robust MDPs. To address this issue, we have added the following paragraph in the related work section to better illustrate the connection between our work and existing literature:
> >     * The uncertainty set captures the discrepancy between training and deployment environments. In our work, we mainly focus on the (s, a)-rectangular uncertainty set (Iyengar, 2005; Nilim \& El Ghaoui, 2004; 2005), where the transition uncertainty is modeled independently for each state-action pair. As shown by Lu et al. (2024); Kumar et al. (2023); Wang et al. (2023a); Wang \& Zou (2021), this structure induced by a specific norm or divergence usually enables efficient solution methods through robust value iteration and linear programming, preserving the tractability of dynamic programming. While our work is also potential to be extended to other settings when a valid robust policy evaluation algorithm presents, including:
> >
> >     * s-Uncertainty Sets: The s-rectangular uncertainty set (Li et al., 2022; Li \& Shapiro, 2023; Kumar et al., 2023) relaxes independence by allowing nature to choose transitions jointly across all actions at a given state. The widely known results include the robust policy gradient formula (Li et al., 2022) and the closed-form solution for p-norm uncertainty sets (Kumar et al., 2023). These results make it possible to solve the robust value function more efficiently.
> >
> >     * d-Uncertainty Sets: More recently, the d-rectangular uncertainty set (Ma et al., 2022) has been proposed for linear MDPs to further generalize the existing concepts of (s, a)-rectangular uncertainty sets, where the ambiguity is structured in a low-dimensional decision space, enabling scalable robust value iteration in high-dimensional settings. Representative works include Blanchet et al. (2023); Liu \& Xu (2024a); Liu et al. (2024); Liu \& Xu (2025). Rmarkably, this setting has also been widely explored in the offline RL setting (Tang et al., 2024; Liu \& Xu, 2024b). These results make it possible to extend our results potentially to be able to handle the robust MDPs with the function approximation These results make it possible to extend our results potentially to be able to handle the constrained robust MDPs with the function approximation
> >
> > * ***Other minor issues***
> >
> >     Thank you for the careful reading. We have addressed all noted minor issues in the revised manuscript.

---

> > ### Comment · Reviewer_WgLD · 2025-07-15
> >
> > Thank you. Most of my questions have been addressed.
> >
> > About your answer to Q3, assumption 4.2. How is it related to this recent ICML 2025 work? It also includes change of measure. I would like the authors to include discussion with this work:
> >
> > Sample Complexity of Distributionally Robust Off-Dynamics Reinforcement Learning with Online Interaction. ICML 2025

---

> > > ### Author Response · Authors · 2025-07-15
> > >
> > > We appreciate the reviewer providing this insightful comment. Except for the difference in the horizon setting, our Assumption 4.2 safeguards a uniform lower bound on the worst-case visitation probability measure, guaranteeing that every state continues to receive a non-negligible share of mass under adversarial dynamics, while instead, [he2025sample] control the relative change of measure via a bounded ratio. The two viewpoints are complementary: if the nominal kernel already satisfies $d^\pi_h(s) \geq p_0 >0$, then our bound implies a finite $C_{vr}$, linking the two frameworks.
> > >
> > > To reflect the connection to this highly related reference, we have updated the manuscript to: (i) include this reference in the introduction section with other robust constrained RL literature, and (ii) add the following discussion right below our Assumption 4.2:
> > >
> > > > Remark: This assumption imposes a uniform lower bound for all policies $\pi$; another widely accepted assumption is made on the finite-horizon scenario [he2025sample], which adopt a complementary requirement that caps the relative re-weighting between nominal and adversarial dynamics.  Exploring algorithms that retain our absolute-coverage guarantee while accommodating the ratio-based perspective of [he2025sample] is an interesting avenue for future work.

---

> ### Comment · Action_Editor_J5un · 2025-07-07
> **Evaluate the authors' response and revision**
>
> Dear Reviewer,
>
> Thank you for submitting your review for the TMLR submission *Rectified Robust Policy Optimization for Robust Constrained Reinforcement Learning without Strong Duality.* The authors have provided a response and submitted a revised version of the paper.
>
> Please take the time to carefully read their response and revision, and update your review accordingly to reflect whether your concerns have been addressed.
>
> Thank you again for your contribution to the review process.
>
> Best,
>
> Pan

---

### Review · Reviewer_F72f · 2025-06-05

**Summary Of Contributions:**

This paper introduce a new algorithm (RRPO) for constrained robust reinforcement learning that addresses the instability of the existing algorithms (CRPO). RRPO introduces a rectification mechanism that stabilizes learning by tracking the best feasible policy. It is shown to be compatible with various uncertainty models (e.g., p-norm and IPM-based) and leverages robust value function approximation methods. The RRPO also enjoys global convergence guarantees, matching known lower bounds for constrained RL.

**Audience:**

Yes

**Broader Impact Concerns:**

None to the best of my knowledge.

**Claims And Evidence:**

Yes

**Requested Changes:**

1.  Add more comprehensive empirical results to substantiate the claim that RRPO improves stability near constraint boundaries.
2.  Provide some theoretical intuition or at least a concrete example illustrating how RRPO’s rectification mechanism can reduce oscillation or improve stability.
3. Include a more detailed comparison with existing constrained RL approaches, especially primal-dual methods.

Minor changes:
1. "We will exchangeably using the parametric..." -> "We will use the parametric and policy representations interchangeably."

**Strengths And Weaknesses:**

Strength: The paper identifies a key weakness in the CRPO framework. The proposed modifications are conceptually simple and compatible with a wide range of robust RL settings and uncertainty models.


Main weakness: The main contribution of the paper lies in the introduction of a threshold tracking mechanism (the $d_0$​ update) and a rule-based triage strategy to switch between objective and constraint optimization based on the threshold gap $\delta$. While this design is motivated by the oscillatory behavior observed in CRPO, the paper does not provide convincing theoretical or empirical evidence that RRPO indeed mitigates this issue. The theoretical analysis only establishes that RRPO achieves the same convergence rate as CRPO. On the empirical side, while RRPO sometimes yields better rewards, the experiments are limited in scope and do not systematically measure oscillation reduction or provide broad benchmarks to support general claims of superiority.

Additionally, the claim that "strong duality does not hold, so we cannot use primal-dual methods" is too strong. Even when strong duality fails, there exist primal-dual approaches that can find approximate stationary points under weaker assumptions such as weak duality. The paper would benefit from a more detailed discussion of this and a clearer justification for ruling out primal-dual approaches entirely.

---

> ### Author Response · Authors · 2025-06-23
>
> We thank the reviewer for the constructive and thoughtful feedback. We have uploaded the revision. Below we provide point-by-point responses.
>
> * ***Q1:*** Some claims are strong.  Even when strong duality fails, there exist primal-dual approaches that can find approximate stationary points under weaker assumptions.
>
>     ***A1:*** Thank you for raising this nuance. Indeed, [Wang2022] can find the stationary point without making the duality assumption. We have softened the wording in Section 1 (Introduction) and have removed related statement in Section 3.2 (Constrained Robust RL Has Non-Zero Duality Gap) to clarify this point and emphasis that existing approaches can find approximate stationary points in the presence of a non-zero duality gap.
>     The revised manuscript has now focused more on the theoretical contribution of constructing the first counterexample with non-zero duality gap in robust constrained RL.
>
>     * [Wang2022] Wang Y, Miao F, Zou S. Robust constrained reinforcement learning[J]. arXiv preprint arXiv:2209.06866, 2022.
>
> * ***Requested Change 1 \& 3:***  Add more comprehensive empirical results \&  Include a more detailed comparison with existing constrained RL approaches, especially primal-dual methods.
>
>     ***Response:*** We sincerely thank the reviewer for their insightful comments and thoughtful suggestions. In response, we have added a robust control experiment in the revised manuscript using the *HalfCheetah-v4* environment from Robust Gymnasium, comparing our RRPO algorithm with robust policy gradient and robust primal-dual baselines. RRPO maintains constraint satisfaction and delivers competitive performance, validating its advantage in settings of robust constrained RL.
>
> * ***Requested Change 2:*** Provide some theoretical intuition or at least a concrete example illustrating how RRPO's rectification mechanism can reduce oscillation or improve stability.
>
>     ***Response:*** Thank you for the insightful comment for a deeper explanation and empirical validation. We would like to clarify that we did not claim that our main contributions include the threshold tracking mechanism in CRPO, as it is a relatively simple trick. Specifically, the original CRPO algorithm does not track the best feasible policy and instead outputs a uniformly sampled policy over the entire trajectory (with a weighted probability). In contrast, our RRPO algorithm introduces a rectification mechanism that "remembers" the highest objective value achieved among feasible policies. This key difference significantly reduces output policy oscillation and improves stability.
>
>     To address the reviewer's request, we have included an ablation study using the robust HalfCheetah-v4 environment. We evaluate the variance of the primal reward of the output policy. In the RRPO algorithm, this variance primarily stems from inherent randomness in the policy and environment. In CRPO, however, an additional source of variance arises from the random sampling of the output policy. As a result, CRPO exhibits substantially higher variance than RRPO.
>
>     We believe these additions provide both theoretical intuition and empirical evidence supporting RRPO's improved stability. We are sincerely grateful for the reviewer's suggestion and hope this enhanced explanation further strengthens the paper.
>
> * ***Minor changes***
>
>      ***Response:*** Thank you for the careful reading. We have addressed all noted minor issues in the revised manuscript.

---

> ### Comment · Action_Editor_J5un · 2025-07-07
> **Evaluate the authors' response and revision**
>
> Dear Reviewer,
>
> Thank you for submitting your review for the TMLR submission *Rectified Robust Policy Optimization for Robust Constrained Reinforcement Learning without Strong Duality.* The authors have provided a response and submitted a revised version of the paper.
>
> Please take the time to carefully read their response and revision, and update your review accordingly to reflect whether your concerns have been addressed.
>
> Thank you again for your contribution to the review process.
>
> Best,
>
> Pan

---

### Review · Reviewer_9eGv · 2025-06-17

**Summary Of Contributions:**

This paper studies the robust constrained reinforcement learning (RL) problem, where an agent must optimize its performance under worst-case transition uncertainties while satisfying safety constraints. The authors prove that strong duality does not generally hold in robust constrained RL—a fact that undermines the efficacy of popular primal-dual algorithms. They provide a counterexample to rigorously demonstrate a non-zero duality gap. Motivated by this, the paper proposes a primal-only algorithm, Rectified Robust Policy Optimization (RRPO), that operates directly on the primal problem and avoids reliance on dual formulations.

**Audience:**

Yes

**Broader Impact Concerns:**

There is no broader impact discussion in the paper.

**Claims And Evidence:**

No

**Requested Changes:**

Please see the weaknesses.

**Strengths And Weaknesses:**

Strengths
- The paper provides the first concrete counterexample demonstrating a non-zero duality gap in robust constrained reinforcement learning. This is a good theoretical contribution that addresses an open problem in the field.
- The proposed Rectified Robust Policy Optimization (RRPO) algorithm is a good solution to the duality gap issue. It avoids the limitations of primal-dual approaches and is supported by rigorous theoretical analysis with convergence guarantees.

Weaknesses
- Insufficient Motivation and Real-World Relevance: The paper merges robust RL and constrained RL but does not clearly articulate the motivation behind this integration or its importance in real-world settings. Without concrete examples or compelling use cases, it is difficult to assess the practical value of addressing this joint formulation. The experiments focus solely on toy environments, making it unclear where and how this approach would be applicable in real-world applications.
- Incomplete Literature Review and Narrow Scope: The related work discussion appears limited, with a noticeable reliance on papers older than five years. Recent developments in both robust and constrained RL over the past three years are largely overlooked. On the robust RL side, the paper focuses exclusively on transition uncertainty, without addressing reward or state uncertainty—raising concerns about the generality of the proposed method. On the constrained RL side, the paper only considers soft constraints expressed via cumulative reward functions, ignoring formulations involving hard safety constraints. Notably, there is no discussion of alternative constraint handling mechanisms such as shielding or model-predictive shielding. As a result, the problem formulation may be too narrow and should be better scoped.
- Limited and Simplistic Experimental Validation: The experimental environments—gridworld and mountain car—are relatively simple and may not adequately demonstrate the effectiveness of the method for complex or realistic settings. Additionally, the baseline comparisons are limited. Given the growing body of work on robust, constrained, and robust-constrained RL, it would strengthen the empirical evaluation to include more recent and competitive baselines from each of these categories.
- Unclear Connection to Deep Reinforcement Learning: Without this counterexample, it is already known that strong duality may not hold for many recent deep reinforcement learning algorithms, particularly in nonconvex settings. This is a common issue in practice. However, the paper does not explain how the proposed RRPO algorithm can be integrated into or improve deep RL methods. Clarifying its applicability to deep RL—and demonstrating this experimentally—would significantly strengthen the paper's practical relevance and impact.

---

> ### Author Response · Authors · 2025-06-23
>
> We thank the reviewer for the constructive and thoughtful feedback. We have uploaded the revision. Below we provide point-by-point responses.
>
> * ***Question:*** Insufficient Motivation and Real-World Relevance
>
>     ***Response:*** Thank you for highlighting the need for clearer motivation. We have revised the Introduction Section and additionally included examples outlining how the same formulation naturally extends to mission-critical finance (robust constrained trading) and healthcare (dose-limited treatment planning), further demonstrating real-world relevance.
>
> * ***Question:*** Incomplete Literature Review and Narrow Scope
>
>     ***Response:*** We appreciate the suggestions to include newer work. We have  revised the Related Work section; it now covers recent advances including hard-constraint mechanisms, safe RL, and robust RL with rewards and state uncertainty.
>
> * ***Question:*** Limited and Simplistic Experimental Validation
>
>     ***Response:*** Thanks for the suggestion. Our initial toy examples were chosen for analytical clarity, while we agree that richer benchmarks strengthen the story. The updated manuscript adds robust control tasks from Robust Gymnasium (the most recent robust RL benchmark) with explicit energy budgets. On these environments RRPO maintains constraint satisfaction and outperform other experimental results. Full learning curves and detailed experiment setting are now included; gridworld and mountain-car results remain as illustrative low-variance sanity checks.
>
> * ***Question:*** Unclear Connection to Deep Reinforcement Learning
>
>     ***Response:***  Thank you for the insightful comment. We agree that in many practical deep RL settings, strong duality may not hold due to nonconvexity introduced by function approximation. Our focus is on a complementary source of duality gap: one that arises fundamentally from the problem formulation itself even in the absence of approximation error.  In other words, our results reveal that a positive duality gap can persist even under perfect function approximation, highlighting a deeper structural limitation. In the revised paper, we have added experiments involving deep neural network approximators to illustrate how our proposed RRPO algorithm can be integrated into deep RL settings.

---

> > ### Comment · Reviewer_9eGv · 2025-07-09
> > **Response to the comments**
> >
> > Thanks for improving the quality of this paper. While the new motivation examples are added, the experiment results cannot support the claim that the proposed methods can solve challenges in either mission-critical finance or healthcare. It is good to see the new literature added, but the discussion between the new literature and the proposed method is missing. For example, the title is quite broad that covering the entire robust RL; however, it does not seem to be able to handle state uncertainties. I would suggest adding the following papers:
> > - What is the Solution for State-Adversarial Multi-Agent Reinforcement Learning?
> > - A Robust and Constrained Multi-Agent Reinforcement Learning Electric Vehicle Rebalancing Method in AMoD Systems

---

> > > ### Author Response · Authors · 2025-07-09
> > >
> > > We thank the reviewer for the insightful comment. We follow the reviewer's suggestion and revise our manuscript as follows:
> > >
> > > * We adjust the statement in describing the connection to real-world applications by emphasizing more on the environment navigation and the robust control task demonstrated in our experiments; the editted statement is shown as follows:
> > >
> > > > For instance, an autonomous robot may encounter unforeseen transitions due to equipment aging or mechanical failures: In the environment navigation and the robust control tasks, the robot make take the action that perform better in the ideal simulated environment due to the ideal equipment condition  but  suffer high penalty from higher energy cost or unexpected operations in the worst-case scenario.
> > >
> > > * We revise the paper title from "robust constrained reinforcement learning" to "model-uncertain constrained reinforcement learning" to better reflect the core focus of our paper.
> > >
> > > * We also thank the reviewer for suggesting these important references. We have updated the manuscript to incorporate them, along with related work in this area. For clarity and ease of review, we provide the revised parts below:
> > >
> > > > ***Robust Constrained Multi-Agent Systems*** An important extension of robust constrained RL is its
> > > application to multi-agent settings. The concept of robustness has been generalized to multi-agent reinforce-
> > > ment learning (MARL) (Zhang et al., 2020; Ma et al., 2023; Jiao \& Li, 2024), particularly in addressing state
> > > uncertainty (He et al., 2023a; Han et al., 2024). Notably, He et al. (2023b) were the first to jointly consider
> > > robustness and constraints in a real-world scenario, proposing the ROCOMA algorithm, which achieves effi-
> > > cient EV rebalancing under transition kernel uncertainty. Many topics in constrained robust MARL remains
> > > unexplored, making it a promising direction with significant potential for further exploration.

---

> ### Comment · Action_Editor_J5un · 2025-07-07
> **Evaluate the authors' response and revision**
>
> Dear Reviewer,
>
> Thank you for submitting your review for the TMLR submission *Rectified Robust Policy Optimization for Robust Constrained Reinforcement Learning without Strong Duality.* The authors have provided a response and submitted a revised version of the paper.
>
> Please take the time to carefully read their response and revision, and update your review accordingly to reflect whether your concerns have been addressed.
>
> Thank you again for your contribution to the review process.
>
> Best,
>
> Pan

---

### Decision · Action_Editor_J5un · 2025-07-29

**Recommendation:** Accept as is

**Additional Comments:**

This paper studies robust constrained reinforcement learning and constructs a specific example to show that strong duality does not generally hold in this setting—an insightful and, to the best of my knowledge, first-of-its-kind contribution in the literature. The authors then propose a primal-only algorithm to solve the robust constrained MDP. Rigorous theoretical analysis and validating experimental results are provided to demonstrate the effectiveness of the proposed algorithm. Through the revision, the authors have also addressed several key limitations of the previous manuscript, including adding discussions of multiple relevant works, incorporating experiments on MuJoCo environments, and clarifying the theoretical assumptions regarding online exploration. Overall, the contributions of this paper—particularly the construction of a hard instance and the development of a theoretically convergent algorithm—are sufficient to merit publication in TMLR.

**Audience:**

Yes

**Audience Explanation:**

Yes, robust reinforcement learning and safe/constrained reinforcement learning are among the most important topics that make RL practically applicable and reliable in real-world applications including healthcare, finance, robotics, etc. This paper will be of interest to many audiences of TMLR.

**Claims And Evidence:**

Yes

**Claims Explanation:**

Yes, the paper designed a specific example to show the existence of non-zero duality gap in constrained robust MDP. An algorithm that does not depend on primal-dual update is proposed and its convergence rate is rigorously proved. Experiments on simulated problems are conducted to validate the theoretical and algorithmic claims.

---

> ### Public Comment · ~Arnob_Ghosh2 · 2025-08-13
> **Related Paper**
>
> This paper seems to be similar to the Epigraph-based paper https://openreview.net/forum?id=G5sPv4KSjR
>
> Like this paper, that paper also considers an epigraph form. However, interestingly, the iteration complexity bound is worse there. Can you explain the difference and why the authors here achieve a better bound?

---

> ### Public Comment · ~Arnob_Ghosh2 · 2025-08-13
> **Related Paper**
>
> This paper seems to be similar to the Epigraph-based paper https://openreview.net/forum?id=G5sPv4KSjR
>
> Like this paper, that paper also considers an epigraph form. However, interestingly, the iteration complexity bound is worse there. Can you explain the difference and why the authors here achieve a better bound?

---

> > ### Author Response · Authors · 2025-08-13
> >
> > Thanks for bringing this paper to our attention; our work is in fact completed earlier than this ICLR paper. As noted in their“Limitations and Future Work” section, the authors acknowledge that the reported iteration complexity may not be tight; for CMDPs, they point out that using the natural policy gradient can improve the complexity to $O(\epsilon^{-2})$. In fact, both in our work and CRPO, the policy updates follow the natural policy gradient, which could be the reason for achieving this improved complexity.

---

> > > ### Public Comment · ~Arnob_Ghosh2 · 2025-08-13
> > > **Thanks for your reply**
> > >
> > > Thanks for your reply. I greatly appreciate this discussion.
> > >
> > > This work was submitted on 10th May (according to the TMLR timeline); the ICLR decisions were out in January, and the ArXiv paper was out before that (https://arxiv.org/abs/2408.16286). So, when you submitted the work, perhaps it was already out. You might not be aware of this.
> > >
> > > I greatly appreciate the non-zero duality gap proof. It is truly a significant contribution.
> > >
> > > However, there are some nuances I did not get in the proof. First, when you are getting (12), you are inherently assuming that $A^{\pi}(s,a)$ is positive. Can you please explain why it is so?
> > >
> > >  The key difference in adapting the CRPO is that, the worst-case probability can be different across $i$s even for the same policy $\pi$. However, this was not reflected in Lemmas C.6, and C.7. In particular, when you are applying Lemma C.6 in Lemma C.6, you can only for the $i$ which has been selected at the iteration $t$, as otherwise the relationship would not be valid.
> > >
> > > Can you please explain these? Thanks a lot..

---

> > > > ### Author Response · Authors · 2025-08-13
> > > >
> > > > Thanks for the information. We indeed missed the arXiv version of this paper.
> > > >
> > > > We obtain equation (12) from Lemma 8 in Zhou et al. (2024) (https://arxiv.org/pdf/2307.08875)
> > > >
> > > > > Ruida Zhou, Tao Liu, Min Cheng, Dileep Kalathil, PR Kumar, and Chao Tian. Natural actor-critic for
> > > > robust reinforcement learning with function approximation. Advances in neural information processing
> > > > systems, 36, 2024.
> > > >
> > > > Regarding the indices, we agree that our notation here is somewhat misleading. In both Lemma C.6 and Lemma C.7, we should use $i_t$ instead of $i$. Our notation follows that of CRPO, where their index $i$ should also depend on $t$.

---

> > > > > ### Public Comment · ~Arnob_Ghosh2 · 2025-08-13
> > > > >
> > > > > Thanks for your reply. My concern is on (12) (not on (11), which you got from Zhou et al.) as Toshinori mentioned [below](https://openreview.net/forum?id=7l63xwAgAW&noteId=GlpQk80KJB)
> > > > >
> > > > > I have further concerns--> In the statement of Theorem 4.4, the paper mentioned $\epsilon_{approx}=\Theta(1/\sqrt{T})$, by carefully looking into in Lemma C.8, it is not clear how the statement of Lemma C.8 is connected to Theorem 4.4.
> > > > >
> > > > > There are typos throughout the proof. For example, in the statement of Lemma C.8, I think that there are typos on the statement $N_0=[ t: V_0^{\pi^*}(\mu)-V_0^{\pi_t}(\mu)\leq d_i-\delta ,\forall i ]$, it seems weird connecting all $i$ with the index $0$.

---

> > > > > > ### Author Response · Authors · 2025-08-26
> > > > > >
> > > > > > We appreciate your valuable suggestions. In response, we have revised the manuscript to address the technical error identified in your comments by introducing an additional assumption that requires the uncertainty set to be sufficiently small. More specifically, we decompose the concerned transition probability in Lemma C.4 using $d_\mu^{\pi',P} = d_\mu^{\pi',P} - d_\mu^{\pi',P'} + d_\mu^{\pi',P'}$. The first two terms will be further upper bounded by the diameter of the uncertainty set.  This decomposition enables us to follow the original proof structure while ensuring correctness, without altering the main results. The revised version is provided in: https://arxiv.org/pdf/2508.17448

---

> ### Public Comment · ~Toshinori_Kitamura1 · 2025-08-13
> **Theoretical concerns**
>
> Hello, I would like to raise several theoretical concerns about this paper.
>
> ---
>
> While the paper states that it considers the general uncertainty setting (page 4), I believe the proof contains a critical issue that can invalidate the claimed result for the general case.
>
> In Lemma C.4, the authors cite the performance difference lemma from [Lemma 8 in Zhou (2024)](https://arxiv.org/pdf/2307.08875). However, I think that the proof of this robust difference lemma implicitly requires (s, a)-rectangularity. Specifically, Lemma 8 of Zhou 2024 proves the result as follows:
>
> $$\\begin{aligned}
> V\_{\\kappa^{\\prime}}^{\\pi^{\\prime}}(\\rho)-V^\\pi(\\rho)
> & =\\mathbb{E}\_{\\kappa^{\\prime}, \\pi^{\\prime}}\\left[\\sum\_{t \\geq 0} \\gamma^t\\left(r\\left(s\_t, a\_t\\right)+\\gamma V^\\pi\\left(s\_{t+1}\\right)-V^\\pi\\left(s\_t\\right)\\right)\\right] \\\\ & \\geq \\frac{1}{1-\\gamma} \\mathbb{E}\_{s \\sim d\_\\rho^{\\pi^{\\prime}, \\kappa^{\\prime}}} \\mathbb{E}\_{a \\sim \\pi^{\\prime}(\\cdot \\mid s)}\\left[A^\\pi(s, a)\\right] .\\end{aligned} \\tag{1}
> $$
>
> The last inequality holds due to the robust Bellman equation under (s, a) rectangularity. While Zhou 2024 does not explicitly describe this step, [page 10 of Li 2022](https://arxiv.org/pdf/2209.10579) clearly explains it.
> If we have (s, a) rectangularity, it holds that
>
> $$
> \\begin{aligned}
> \\mathbb{E}\_{\\kappa^{\\prime}, \\pi^{\\prime}}\\left[r\\left(s\_t, a\_t\\right)+\\gamma V^\\pi\\left(s\_{t+1}\\right)\\right]
> &=
> \\mathbb{E}\_{\\kappa^{\\prime}, \\pi^{\\prime}}\\left[r\\left(s\_t, a\_t\\right)+\\gamma \\sum\_{s'} P'(s' \\mid s\_t, a\_t) V^\\pi(s')\\right] \\\\
> &\\geq
> \\mathbb{E}\_{\\kappa^{\\prime}, \\pi^{\\prime}}\\left[r\\left(s\_t, a\_t\\right)+\\gamma \\min\_{p \\in \\mathcal{P}\_{s\_t, a\_t}} \\sum\_{s'} p(s')V^\\pi(s')\\right] \\\\
> & = \\mathbb{E}\_{\\kappa^{\\prime}, \\pi^{\\prime}}\\left[Q^{\\pi}(s\_t, a\_t) \\right]
> \\end{aligned}
> $$
>
> which leads to the above Equation (1).  Note that [Zhou 2024](https://arxiv.org/pdf/2307.08875) assumes (s, a) rectangulairty, so this step holds.  However, without rectangularity, we cannot take $\\min_{p \\in \\mathcal{P}_{s_t, a_t}}$ in this way, and thus the robust value function $Q^\\pi$ may not appear at the right-hand side.
>
> ---
>
> Additionally, as [pointed out above](https://openreview.net/forum?id=7l63xwAgAW&noteId=o3MOn1ZkMM), I think there is no guarantee that Equation (12) holds. Equation (11) and (12) state that:
>
> $$
> \\frac{1}{1-\\gamma} \\mathbb{E}\_{s \\sim d\_\\mu^{\\pi^{\\prime}, P^{\\prime}}} \\mathbb{E}\_{a \\sim \\pi^{\\prime}(\\cdot \\mid s)}\\left[A^\\pi(s, a)\\right] \\leq V^{\\pi^{\\prime}}(\\mu)-V^\\pi(\\mu) \\leq \\frac{1}{1-\\gamma} \\mathbb{E}\_{s \\sim d\_\\mu^{\\pi^{\\prime}, P}} \\mathbb{E}\_{a \\sim \\pi^{\\prime}(\\cdot \\mid s)}\\left[A^\\pi(s, a)\\right] .\\tag{11}
> $$
> $$
> \\frac{C\_{\\ell}}{1-\\gamma} \\mathbb{E}\_{(s, a) \\sim d\_\\mu^{\\pi^{\\prime}, P^{\\prime}} \\otimes \\pi^{\\prime}}\\left[A^\\pi(s, a)\\right] \\leq V^{\\pi^{\\prime}}(\\mu)-V^\\pi(\\mu) \\leq \\frac{C\_u}{1-\\gamma} \\frac{1}{1-\\gamma} \\mathbb{E}\_{(s, a) \\sim d\_\\mu^{\\pi^{\\prime}, P^{\\prime}} \\otimes \\pi^{\\prime}}\\left[A^\\pi(s, a)\\right] \\tag{12}
> $$
>
> However, (12) does not hold when $\\mathbb{E}\_{a \\sim \\pi^{\\prime}(\\cdot \\mid s)}\\left[A^\\pi(s, a)\\right] \leq 0$, which can clearly occur if $\\pi'$ is chosen to be greedy with respect to $-A^\\pi(s, a)$.
> In such cases, the inequalities in (12) may fail.
>
> Moreover, the authors state that (12) follows from Lemma 8 in Zhou (2024), but Zhou (2024) only proves (11), not (12).
> If Equation (12) is incorrect, the convergence result may not hold even under (s, a)-rectangularity.
>
> ---
>
> Overall, I am somewhat skeptical that the proofs have been thoroughly verified.
> Please let me know if I am misunderstanding something.

---

> ### Public Comment · ~Arnob_Ghosh2 · 2025-08-27
>
> I thank the authors for acknowledging the mistake and rectifying it by adding an Assumption. I think that the proofs should go through now. I hope that the discussion will be added to the camera-ready version.
>
> I do have another comment. With the new assumption, the algorithm only proves that there is always a **constant** gap depending on $c$, and cannot be closed for any $\epsilon<c$. Now, we can take any algorithm that solves the CMDP up to $\epsilon$ gap, then using Assumption 4.4, one can show that the policy will incur additional cost of $O(c/(1-\gamma))$ if only learnt using the nominal model, which is also discussed in Theorem 4.4 in https://proceedings.mlr.press/v151/panaganti22a/panaganti22a.pdf.  Hence, the algorithm that solves CMDP (and thus, the CRPO) should have the same bound. If it is so, I would suggest including the discussion as well.

---

> > ### Author Response · Authors · 2025-08-28
> >
> > Thanks for the careful check confirming that the proof now goes through and for the insightful suggestions. We will add a discussion titled "Discussion on the correction from the previous version of this work" under Remark 4.6 to explicitly state the constant gap caused by Lemma C.5, refer it to the related discussion in this thread, and note that the same bound potentially obtained from existing CMDP results as implied by the Panaganti's result.

---

> > > ### Public Comment · ~Toshinori_Kitamura1 · 2025-08-30
> > > **Theoretical concerns (Cont.)**
> > >
> > > Thank you for revising the paper. Could you answer my question regarding the rectangularity of the robust performance difference lemma? (
> > > https://openreview.net/forum?id=7l63xwAgAW&noteId=GlpQk80KJB)
> > >
> > > From my reading, the new paper still considers the general uncertainty set but does not explain how C.5 holds in this case. I again remark that Zhou et al (2024) assume rectangularity.
> > >
> > > I think this lemma is very important to establish $\varepsilon^{-2}$ complexity. Indeed, it is the reason why I could not use the natural policy gradient and ended up with $\varepsilon^{-4}$ complexity in my RCMDP paper: https://arxiv.org/pdf/2408.16286 (I slightly discuss this problem around Eq.(12) of my RCMDP paper). Specifically, **even if we assume (s, a)-rectangularity**, the epigraph form faces the following non-rectangular optimization problem in its constraint:
> > >
> > > $$
> > > \\max \_{d\_0} d\_0
> > > \\text { s.t. } \\max\_{\\pi} \\min\_i \\{V\_i^\\pi(\\mu) - d\_i\\} \\geq 0
> > > $$
> > >
> > > Note that $\\max\_{\\pi} \\min\_i \\{V\_i^\\pi(\\mu) - d\_i\\}$ is not a $(s, a)$-rectangular problem. Due to this non-rectangularity, I was unable to apply the difference lemma and avoided using the natural policy gradient method.
> > >
> > > Thus, I believe that establishing the difference lemma without assuming rectangularity is the key technical challenge for achieving $\varepsilon^{-2}$ complexity with the epigraph formulation. However, I still do not see how your paper addresses this challenge.